# Trust It or Not: Evidential Uncertainty for Feed-Forward 3D Reconstruction with Trust3R

**Zihao Zhu** [* 1] **Wenyuan Zhao** [* 1] **Nuo Chen** [1] **Chao Tian** [† 1] **Zhiwen Fan** [† 1]

## Abstract

Geometric foundation models hold promise for unconstrained dense geometry prediction from uncalibrated images. However, current feed-forward designs often produce heuristic confidence scores that lack probabilistic interpretation and fail to indicate where and how much the predicted geometry can be trusted. To address this gap, we present *Trust3R*, a lightweight evidential uncertainty framework for feed-forward 3D reconstruction. Trust3R combines gated residual mean refinement with a Normal-Inverse-Wishart evidential head, yielding a closed-form multivariate Student-$t$ distribution for per-point geometric uncertainty. This provides probabilistically grounded pointmap uncertainty estimates with moderate inference overhead. We evaluate on diverse indoor and outdoor benchmarks and compare against MASt3R's built-in confidence map, single-pass heteroscedastic regression, MC dropout, and deep ensembles. Experimental results show that Trust3R consistently improves risk–coverage and sparsification, generally improves geometric accuracy, and strengthens uncertainty ranking across benchmarks. On ScanNet++, Trust3R achieves 25% lower AURC and 41% lower AUSE, providing a practical reliability signal for uncertainty-aware weighting in downstream geometry pipelines. Our project page and code are available at https://trust3r-z.github.io/.

## 1. Introduction

Recent advancements in feed-forward 3D reconstruction have significantly reshaped the way geometric reasoning is

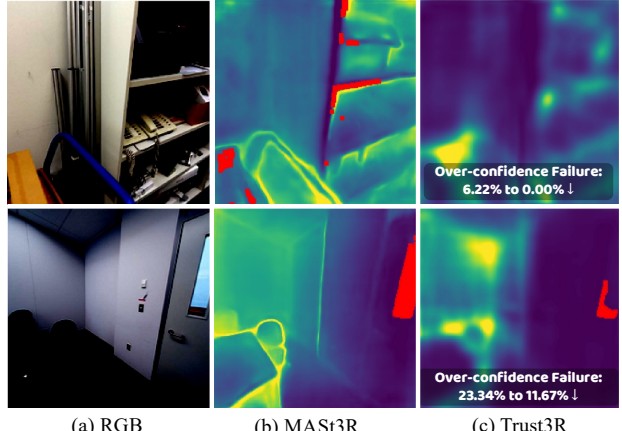

(a) RGB      (b) MASt3R      (c) Trust3R

*Figure 1.* **Reducing overconfident geometric failures with evidential uncertainty map**: the 3D geometric pixels, which falls into the top-$q_{\mathrm{err}}\%$ reconstruction errors while being assigned the lowest $q_{\mathrm{unc}}\%$ uncertainty, are highlighted in red as overconfident failures. In contrast to the heuristic confidence in MASt3R, Trust3R reduces these incorrect yet highly confident regions, resulting in uncertainty estimates that better align with empirical reconstruction errors and can be used for downstream weighting.

performed from images (Wang et al., 2024; Leroy et al., 2024). Instead of explicitly estimating camera parameters and enforcing multi-view consistency through iterative optimization, modern approaches directly regress dense 3D representations from image pairs or collections (Yao et al., 2018; Sun et al., 2021). DUSt3R (Wang et al., 2024)/MASt3R (Leroy et al., 2024) established a particularly influential paradigm by grounding image matching directly in 3D: given a pair of images, the model predicts dense pointmaps whose geometric consistency implicitly encodes correspondence and depth. Beyond isolated reconstruction tasks, such feed-forward geometric priors are increasingly being adopted as systems-level building blocks, driving recent advances in real-time mapping systems (Murai et al., 2025; Maggio et al., 2026), robotic systems (Lin et al., 2025; Yu et al., 2026), and geometry-aware world models (Wu et al., 2025; Qian et al., 2025; Sun et al., 2026).

However, feed-forward 3D reconstruction remains fundamentally ambiguous: occlusions, repetitive structures, low-texture regions, distribution shifts, and viewpoint degeneracies often admit multiple geometrically valid explana-

---
[*]Equal contribution   [†]Equal advising. [1] Department of Electrical and Computer Engineering, Texas A&M University, College Station, TX, USA . Correspondence to: Zhiwen Fan <zhiwenfan@tamu.edu>.

*Proceedings of the $43^{rd}$ International Conference on Machine Learning*, Seoul, South Korea. PMLR 306, 2026. Copyright 2026 by the author(s).

tions (Wang et al., 2025a; Keetha et al., 2026; Wang et al., 2025b; Wang & Agapito, 2024). This ambiguity makes confidence assessment essential: beyond plausible geometry, we need predictions that indicate *where* they may fail. Accordingly, the focus is shifting from merely producing plausible 3D geometry to estimating its reliability (Lakshminarayanan et al., 2017; Kendall et al., 2018; Sattler et al., 2019; Besnier et al., 2021). Although MASt3R improves robustness compared to previous methods, it still predicts a single *deterministic* pointmap together with a per-pixel confidence score that acts as a learned reliability weight for the regression loss, rather than a predictive uncertainty with a clear probabilistic interpretation. As a result, confidence maps may not faithfully capture where and how much the geometry is reliable.

On the other hand, uncertainty quantification (UQ) offers a principled way to address this limitation (Blundell et al., 2015; Gal & Ghahramani, 2016; Lakshminarayanan et al., 2017; Kendall et al., 2018; Amini et al., 2020), but many widely used UQ paradigms rely on Bayesian sampling (Janssen, 2013) model inference, which becomes prohibitive for dense 3D pointmaps and conflicts with the real-time requirements of many real-world applications. Motivated by this, *evidence-based* uncertainty modeling offers a feed-forward alternative for uncertainty-aware 3D reconstruction: it directly predicts the parameters of a predictive distribution together with an explicit notion of evidence supporting the prediction, yielding closed-form uncertainty estimates with a clear probabilistic interpretation and no need for sampling, as in deep evidential learning (Sensoy et al., 2018). However, evidential learning is largely developed for low-dimensional regression with (approximately) independent outputs, and it is not plug-and-play for dense pointmap prediction. Vanilla evidential regression is designed for low-dimensional, independent multitasking regression outputs (Soleimany et al., 2021), whereas feed-forward pointmap reconstruction involves hundreds of thousands of correlated 3D points per image pair subject to strong geometric constraints and strict efficiency demands (Scharstein & Szeliski, 2002; Wang et al., 2024). Naively porting evidential heads to this setting can lead to unstable training and poorly estimated uncertainty with empirical geometric errors. This calls for a careful refinement strategy that preserves both the predictive accuracy and reliable uncertainty estimation.

To address the lack of reliable per-point uncertainty estimation in dense feed-forward pointmap prediction, we design **Trust3R**, a lightweight evidential uncertainty framework for feed-forward pointmap reconstruction. First, Trust3R augments feed-forward multi-view geometry prediction with probabilistic outputs and an uncertainty estimate for each 3D point via a lightweight evidential UQ head, enabling the model to estimate which regions are more likely to

be geometrically unreliable, learned from data. Second, Trust3R introduces a gated refinement module that updates predictions through a residual connection, preserving the geometric accuracy and generalization of pretrained geometric foundation models while stabilizing evidential learning under challenging conditions such as occlusions and distribution shifts. As shown in Figure 1, Trust3R produces efficient yet reliable uncertainty estimates that reduce overconfident geometric failures, making uncertainty usable as a reliability signal for filtering and weighting in downstream geometry. We summarize the following contributions:

- We introduce Trust3R, enabling geometric foundation models to produce probabilistically grounded uncertainty estimates for dense pointmaps via evidential learning. This supports explicit evaluation of geometric reliability under ambiguity and distribution shifts.

- Trust3R introduces pointmap-tailored designs for end-to-end evidential learning, combining an evidence-regularized predictive distribution with a gated residual refinement when needed. This preserves the strength of pretrained geometric foundation models while preventing overconfident uncertainty.

- We demonstrate the effectiveness of our predictive distribution and uncertainty estimates across a range of 3D perception benchmarks. Our method achieves a strong trade-off between single-pass inference efficiency and UQ quality while enabling practical reliability-aware filtering, alignment, and fusion.

## 2. Related Work

**3D geometric foundation models.** Recent feed-forward 3D models regress geometry directly from images using large transformers, including depth- and correspondence-based approaches (Luo et al., 2016; Ranftl et al., 2021; Birkl et al., 2023) and pointmap formulations (Qian et al., 2022; Yu et al., 2022). DUSt3R (Wang et al., 2024) predicts dense pairwise 3D pointmaps in a shared coordinate frame, while MASt3R (Leroy et al., 2024) improves robustness via mask-aware training and refined heads, achieving strong accuracy without iterative optimization. Pointmap representations offer an efficient and expressive backbone to downstream optimization modules, including Structure-from-Motion and visual SLAM systems (Schonberger & Frahm, 2016; Zhou et al., 2016; Dellaert & Kaess, 2017).

**UQ in 3D geometry.** Trust-aware 3D reconstruction has relied on heuristic confidence scores alongside dense pointmap predictions, but these scores lack a probabilistic interpretation and are often poorly estimated (Wang et al., 2024; Leroy et al., 2024; Kersting et al., 2007; Blundell et al., 2015). Bayesian approaches, including MC dropout (Gal & Ghahramani, 2016) and deep ensembles (Lakshminarayanan et al., 2017), offer uncertainty estimates by marginalizing

model likelihood, yet they rely on multiple stochastic forward passes or multiple trained models (Carlin & Louis, 2008; Kruschke, 2010; Wang & Yeung, 2016). For dense 3D pointmap prediction with hundreds of thousands of correlated outputs per image pair, this sampling-based paradigm introduces prohibitive computational overhead. Conformal prediction (Shafer & Vovk, 2008; Angelopoulos et al., 2023) provides distribution-free uncertainty guarantees, but its worst-case nature often leads to overly conservative uncertainty regions and does not align with the fine-grained per-point 3D geometry (Mo et al., 2019; Fillioux et al., 2024). These limitations motivate evidential learning as a promising alternative (Sensoy et al., 2018; Amini et al., 2020; Gao et al., 2025): by predicting distributional parameters in a single forward pass, evidential methods offer probabilistically grounded uncertainty estimates with negligible inference overhead, making them particularly attractive for large-scale feed-forward 3D reconstruction.

**Multivariate evidential regression.** Multivariate deep evidential regression (Meinert & Lavin, 2021) extends evidential regression to multivariate outputs by using a Normal-Inverse-Wishart prior and the induced multivariate Student-$t$ predictive distribution. We adopt this probabilistic formulation for dense feed-forward pointmap uncertainty estimation, where the goal is to predict uncertainty for spatially organized 3D outputs rather than low-dimensional regression targets.

# 3. Methodology

We consider the problem of dense 3D reconstruction from a set of images. Given one or more input images $\mathcal{I} = \{\mathcal{I}^v\}_{v=1}^N$, a 3D reconstruction model predicts a dense 3D point-cloud, denoted as *pointmap*, $\mathbf{X} = \{\mathbf{X}_i \in \mathbb{R}^3\}_{i=1}^M$, where each $\mathbf{X}_i$ represents a 3D point associated with a pixel or patch in the input images. In existing pointmap-based methods, such as DUSt3R and MASt3R, $\mathbf{X}$ is treated as a deterministic output of a neural network. Figure 2 overviews our uncertainty-aware pipeline.

### 3.1. Preliminary

DUSt3R and MASt3R formulate pairwise reconstruction as a feed-forward prediction problem. Given an image pair $(\mathcal{I}^1, \mathcal{I}^2)$, the network predicts two dense pointmaps (in a chosen reference frame) together with auxiliary maps used for matching and filtering

$$(\mathbf{X}^1, \mathbf{c}^1), \ (\mathbf{X}^2, \mathbf{c}^2) = f_{\text{pair}}(\mathcal{I}^1, \mathcal{I}^2), \tag{1}$$

where $\mathbf{X}^v \in \mathbb{R}^{H \times W \times 3}$ is the per-pixel pointmap prediction for view $v \in \{1, 2\}$, and $\mathbf{c}^v \in [0, 1]^{H \times W}$ is a learned confidence/validity score.

Importantly, the confidence in DUSt3R/MASt3R is *not* a predictive uncertainty with a probabilistic interpretation

(e.g., variance of a likelihood). Instead, it is a learned scalar that serves as a heuristic reliability indicator, typically used to (i) down-weight or filter unreliable pixels during training and/or (ii) select pixels for downstream alignment and fusion. As a consequence, such scores do not naturally support uncertainty propagation across views and can be poorly aligned with geometric errors under occlusions, textureless/reflective regions, and distribution shifts. This motivates replacing heuristic confidence with a predictive distribution whose uncertainty can be interpreted and consumed by downstream geometry modules.

### 3.2. Uncertainty-aware Pointmap Reconstruction

While MASt3R refines pointmap regression through geometrical and feature matching, the remaining valid pixels are still deterministic predictions but exhibit varying degrees of uncertainty due to noise, limited visual evidence, and inherent ambiguity in pairwise reconstruction.

For these reasons, we propose to substitute the deterministic pointmap 3D head with an *uncertainty-aware evidential head* that models each pointmap as a distribution rather than a single estimate. Specifically, for each pixel $i$ and view $v$, the UQ head predicts a conditional distribution, $p(\mathbf{X}_i^v \mid \mathcal{I})$, capturing both the predicted pointmap and its associated uncertainty.

We assume that every 3D point $\mathbf{X}_i$ is drawn from a multivariate Gaussian likelihood

$$\mathbf{X}_i \sim \mathcal{N}(\boldsymbol{\mu}_i, \boldsymbol{\Sigma}_i), \quad i \in \Omega, \tag{2}$$

where $\boldsymbol{\mu}_i \in \mathbb{R}^3$ is the mean of 3D points and $\boldsymbol{\Sigma}_i \in \mathbb{R}^{3 \times 3}$ is a covariance matrix encoding spatial uncertainty, and $\Omega$ denotes the set of all 3D points. Rather than predicting $(\boldsymbol{\mu}_i, \boldsymbol{\Sigma}_i)$ directly, we adopt evidential learning in which the network predicts *distributional parameters* governing a family of predictive distributions.

### 3.3. Evidential UQ Head and Loss

To probabilistically estimate uncertainty over $(\boldsymbol{\mu}, \boldsymbol{\Sigma})$, we place a conjugate prior over these parameters. Assuming the multivariate Gaussian likelihood, this leads to a Normal–Inverse–Wishart (NIW) prior over the unknown mean and covariance:

$$\boldsymbol{\mu} \mid \boldsymbol{\Sigma} \sim \mathcal{N}(\mathbf{m}, \boldsymbol{\Sigma}/\kappa), \tag{3}$$

$$\boldsymbol{\Sigma} \sim \mathcal{W}^{-1}(\boldsymbol{\Psi}, \nu), \tag{4}$$

where $\mathbf{m} \in \mathbb{R}^3$ is the prior mean, $\kappa > 0$ controls the strength of belief in $\mathbf{m}$. $\mathcal{W}^{-1}(\cdot)$ is the inverse Wishart distribution, $\boldsymbol{\Psi} \in \mathbb{R}^{3 \times 3}$ is a positive-definite scale matrix, and $\nu > d + 1$ ($d = 3$) denotes the degrees of freedom. We denote the set of *evidential* parameters by

$$\boldsymbol{\theta} := \{\mathbf{m}, \kappa, \boldsymbol{\Psi}, \nu\}. \tag{5}$$

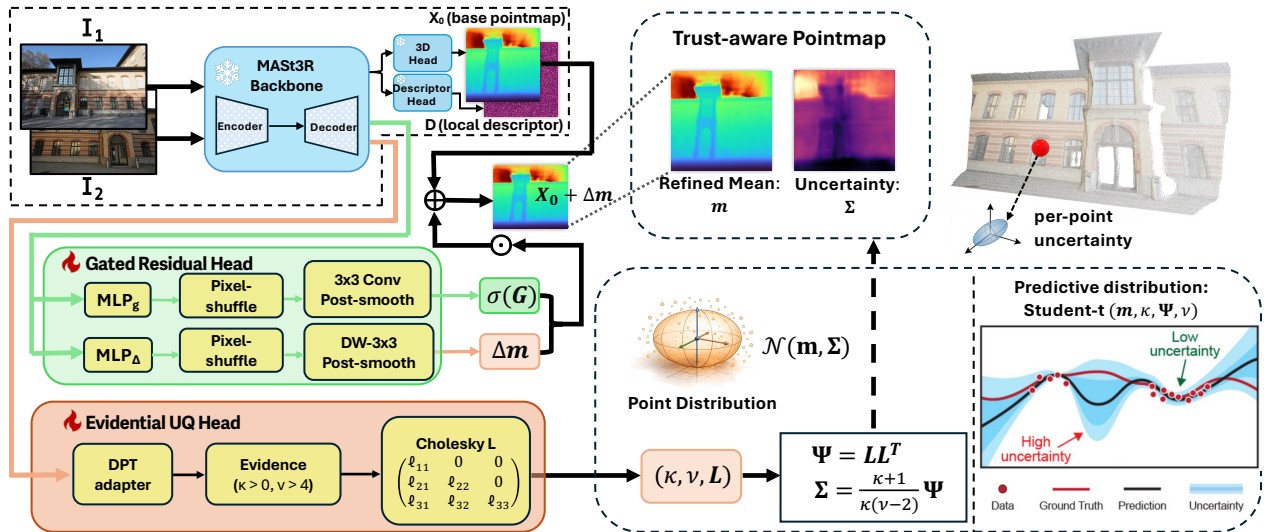

*Figure 2.* **Overview of the proposed Trust3R framework.** Built on a feed-forward MASt3R backbone, we augment pointmap prediction with a gated residual head for geometry refinement and an evidential UQ head that predicts the uncertainty-aware pointmap. This yields a closed-form multivariate Student-$t$ predictive distribution, enabling per-point uncertainty estimation with negligible inference overhead. The evidential uncertainty can be used for reliability evaluation and uncertainty-weighted downstream geometry modules.

Under this formulation, the posterior distribution takes the form of the NIW distribution with joint density:

$$p(\boldsymbol{\mu}, \boldsymbol{\Sigma} \mid \boldsymbol{\theta}) = \mathcal{N}\left(\boldsymbol{\mu} \mid \mathbf{m}, \frac{\boldsymbol{\Sigma}}{\kappa}\right) \mathcal{W}^{-1}(\boldsymbol{\Sigma} \mid \boldsymbol{\Psi}, \nu). \quad (6)$$

From Bayesian probabilistic theory, the "model evidence" (or marginal likelihood) is obtained by integrating out the Gaussian likelihood parameters $(\boldsymbol{\mu}, \boldsymbol{\Sigma})$, which yields a multivariate Student-$t$ predictive distribution over the pointmap:

$$p(\mathbf{X} \mid \boldsymbol{\theta}) = \mathrm{St}\left(\mathbf{X} \mid \mathbf{m}, \frac{\boldsymbol{\Psi}(\kappa + 1)}{\kappa(\nu - 2)}, \nu - 2\right). \quad (7)$$

**Evidential UQ head.** Let $\mathbf{H}^v$ and $\mathbf{H}'^v$ denote the embedded representations produced by the encoder and decoder of MASt3R's ViT (Dosovitskiy et al., 2020), respectively. For each pixel $i \in \Omega$ and view $v \in \{1, 2\}$, the UQ head predicts the uncertainty parameters of a NIW distribution on the predicted point $\mathbf{X}_i^v$:

$$\{\kappa_i^v, \mathbf{L}_i^v, \nu_i^v\} = \mathrm{Head}_{\mathrm{UQ}}^v([\mathbf{H}^v, \mathbf{H}'^v]), \quad (8)$$

where $\mathbf{L}_i^v$ is a lower-triangular factor defined as the Cholesky decomposition of the positive-definite matrix $\boldsymbol{\Psi}_i^v$. Instead of directly using the predictive mean $\mathbf{m}_i^v$ as output, we design a gated residual head to fine-tune the prediction from the pre-trained MASt3R backbone, which is deferred to Section 3.4 in detail.

The objective of the UQ head is to predict the distribution $p(\mathbf{X}^v \mid \mathcal{I}^v)$ of the whole pointmap. Following the mean-field assumption across predicted 3D points, we predict a variational distribution of independent 3D points

$$q(\boldsymbol{\mu}^v, \boldsymbol{\Sigma}^v \mid \boldsymbol{\theta}^v) = \prod_{i \in \Omega} p(\boldsymbol{\mu}_i^v, \boldsymbol{\Sigma}_i^v \mid \boldsymbol{\theta}_i^v), \quad (9)$$

parameterized by the UQ head in Eq. (8), without explicitly observing multiple samples per pixel. This factorization captures cross-coordinate covariance within each 3D point, but does not explicitly model cross-pixel covariance; modeling dense spatial uncertainty is left for future work.

**Evidential UQ loss.** Given the ground-truth 3D points $\hat{\mathbf{X}}_i^{v,1}$, training is performed by minimizing the negative log-likelihood (NLL) of the marginal predictive distribution in Eq. (7):

$$\mathcal{L}_{\mathrm{NLL}} = - \sum_{v \in \{1,2\}} \sum_{j \in \Omega} \log p(\hat{\mathbf{X}}_j^{v,1} \mid \boldsymbol{\theta}_j^v). \quad (10)$$

Minimizing the NLL alone admits degenerate solutions in which the model expresses unjustified certainty by inflating evidence parameters. To mitigate this problem, we introduce an evidence regularizer that penalizes excessive confidence when prediction errors are large.

Following the evidential learning framework, we define the total *evidence* for pixel $i$ as $e_i = \kappa_i + \nu_i$, which jointly controls the concentration of the NIW distribution. The regularization term that penalizes high evidence under large prediction errors is defined as:

$$\mathcal{L}_{\mathrm{evi}} = \sum_{v \in \{1,2\}} \sum_{i \in \Omega} \|\hat{\mathbf{X}}_i^{v,1} - \mathbf{m}_i^v\|^2 \cdot (\kappa_i^v + \nu_i^v). \quad (11)$$

The final evidential UQ loss for uncertainty-aware pointmap regression is given by

$$\mathcal{L}_{\text{UQ}} = \mathcal{L}_{\text{NLL}} + \lambda_{\text{evi}}\mathcal{L}_{\text{evi}}, \tag{12}$$

where $\lambda_{\text{evi}}$ controls the strength of the evidence regularization. In practice, the overall training objective combines the base regression loss with $\mathcal{L}_{\text{UQ}}$ (via a convex mixing for stability). To avoid patch-grid artifacts introduced by token-to-pixel upsampling in the residual branch, we apply an identity-initialized post-upsampling smoothing module (Section A.3), and optionally regularize the spatial variation of the gating map with a weak total-variation prior.

### 3.4. Evidential Gate and Residual Head

The evidential NLL loss requires an accurate predictive mean to ensure stable optimization and avoid degenerate uncertainty (e.g., trivially inflating variance to reduce NLL), which is crucial for reliable ranking under risk–coverage and sparsification. However, directly fine-tuning the mean prediction of a large pre-trained MASt3R model risks degrading its geometric accuracy and losing the benefits of pretraining. To preserve the original pointmap predictions while enabling uncertainty-aware learning, we seek a lightweight refinement mechanism that minimally perturbs the pretrained model.

**Gated residual head.** Let $\mathbf{X}_0^v \in \mathbb{R}^{H \times W \times 3}$ denote the pointmap predicted by the frozen MASt3R backbone for view $v$. Instead of modifying this prediction directly, we introduce a gated residual head that predicts a residual correction:

$$\Delta\mathbf{m}^v, \mathbf{G}^v = \text{Head}_{\text{GR}}^v([\mathbf{H}^v, \mathbf{H}'^v]), \tag{13}$$

along with a gating map $\sigma(\mathbf{G}^v) \in [0,1]^{H \times W}$, which modulates the influence of the residual. The refined predictive mean is then given by

$$\mathbf{m}^v = \mathbf{X}_0^v + \sigma(\mathbf{G}^v) \odot \Delta\mathbf{m}^v, \tag{14}$$

where $\odot$ denotes element-wise multiplication broadcast over the 3D coordinates.

The gating mechanism allows the model to selectively adjust the pretrained mean only where necessary, while leaving confident and well-predicted regions unchanged. When $\sigma(\mathbf{G}^v) \approx 0$, the refined mean reduces to the original MASt3R prediction; when $\mathbf{G}^v$ has large positive logits (so $\sigma(\mathbf{G}^v) \approx 1$), the model is free to apply the predicted residual correction. This design ensures stable fine-tuning and prevents large deviations from the pretrained geometry.

To suppress patch-grid artifacts caused by token-to-pixel upsampling in the residual branch, we further apply a lightweight depthwise-separable post-smoothing operator after upsampling, initialized as an identity mapping so that the refinement starts from an exact no-op.

### 3.5. Predictive Mean and Uncertainty

In contrast to heuristic confidence maps, which assign only scalar scores that often lack a probabilistic interpretation, the evidential UQ head represents each pointmap as a full predictive distribution with well-defined uncertainty semantics. As a result, evidential uncertainty across views can be estimated analytically via Bayesian updates, rather than ad-hoc thresholding or reweighting.

The evidential formulation enables a principled decomposition of uncertainty: *aleatoric* and *epistemic*.

$$\underbrace{\mathbb{E}[\boldsymbol{\mu}] = \mathbf{m}}_{\text{prediction}}, \ \underbrace{\mathbb{E}[\boldsymbol{\Sigma}] = \frac{\boldsymbol{\Psi}}{\nu - 4}}_{\text{aleatoric}}, \ \underbrace{\text{Var}[\boldsymbol{\mu}] = \frac{\boldsymbol{\Psi}}{\kappa(\nu - 4)}}_{\text{epistemic}}. \tag{15}$$

Aleatoric uncertainty arises from observational noise and inherent ambiguity. Epistemic uncertainty is encoded through the evidence parameters $(\kappa, \nu)$: low values correspond to weak evidence and higher model uncertainty, while large values indicate confident predictions supported by strong evidence. This distinction is particularly important in 3D reconstruction, where ambiguity may stem either from intrinsic scene properties (e.g., textureless regions) or from limited training coverage. This property allows pointmap uncertainty to be propagated and combined consistently in downstream inference modules, rather than being heuristically reweighted or thresholded.

Importantly, this formulation also yields uncertainty estimates in closed form and does not require Monte Carlo sampling at inference time. Each reconstructed point is associated with a compact set of evidential parameters, enabling efficient propagation of uncertainty through downstream geometric operations such as alignment, fusion, and filtering. This makes evidential pointmap reconstruction particularly suitable for large-scale and real-time 3D systems.

## 4. Experiments

In this section, we present the evaluation of Trust3R on indoor/outdoor benchmarks. We examine both uncertainty ranking quality and reconstruction accuracy, and report efficiency comparisons to quantify compute overhead. We further provide ablation studies to isolate the effects of key design choices, and include qualitative visualizations to illustrate typical failure modes. Additional experimental results including probabilistic scoring results (NLL) are deferred to the appendix.

### 4.1. Experimental Setup

For each image, we compute per-pixel 3D point errors and associate each pixel with a scalar uncertainty score. Unless noted otherwise, we apply a per-image Sim(3) alignment

*Table 1.* **Uncertainty evaluation** on ScanNet++, TUM RGB-D, and KITTI, computed over valid pixels. We report AURC, AUSE, and Spearman $\rho$. Single-pass uses one inference pass, while multi-pass uses $T$ stochastic inference passes or $K$ trained models. Time reports training time in hours (h). Avg. reports the average measurement across the three datasets. **Bold** and underline mark the best and second-best results within the single-pass block.

| Inference | Method | Time | Avg. | | | ScanNet++ (Indoor) | | | TUM RGB-D (Indoor) | | | KITTI (Outdoor) | | |
|---|---|---|---|---|---|---|---|---|---|---|---|---|---|---|
| | | | AURC↓ | AUSE↓ | $\rho$↑ | AURC↓ | AUSE↓ | $\rho$↑ | AURC↓ | AUSE↓ | $\rho$↑ | AURC↓ | AUSE↓ | $\rho$↑ |
| Sampling | MCD ($T$=16) | – | 0.4902 | 0.2726 | 0.2644 | 0.1777 | 0.0989 | 0.1937 | 0.0672 | 0.0368 | 0.2747 | 1.2257 | 0.6821 | 0.3249 |
| | DeepEns ($K$=5) | 100.2 | 0.2992 | 0.0916 | 0.4556 | 0.0722 | 0.0381 | 0.3921 | 0.0453 | 0.0208 | 0.3127 | 0.7802 | 0.2159 | 0.6620 |
| Single-pass | MASt3R | – | 0.4155 | 0.2053 | 0.4057 | 0.1649 | 0.0747 | 0.2837 | 0.0538 | 0.0233 | 0.4812 | 1.0277 | 0.5178 | 0.4523 |
| | Hetero | 9.4 | 0.3973 | 0.1872 | 0.4183 | 0.1616 | 0.0715 | 0.3545 | 0.0555 | 0.0251 | 0.4669 | **0.9749** | 0.4650 | 0.4335 |
| | **Trust3R** (ours) | 10.5 | **0.3861** | **0.1684** | **0.4898** | **0.1233** | **0.0444** | **0.4930** | **0.0481** | **0.0178** | **0.5169** | 0.9868 | **0.4431** | **0.4596** |

*Table 2.* **Reconstruction accuracy** measured by pointmap 3D point error. We report MAE and RMSE on ScanNet++, TUM RGB-D, and KITTI, computed over valid pixels after Sim(3) alignment. **Bold** marks the best result in each column; smaller errors are better.

| Method | ScanNet++ | | TUM RGB-D | | KITTI | |
|---|---|---|---|---|---|---|
| | MAE↓ | RMSE↓ | MAE↓ | RMSE↓ | MAE↓ | RMSE↓ |
| MASt3R | 0.2164 | 0.3026 | 0.0938 | 0.1600 | **1.6108** | **3.0427** |
| **Trust3R** | **0.1959** | **0.2849** | **0.0873** | **0.1496** | 1.6648 | 3.0772 |

*Table 3.* **Computational overhead.** We report training wall-clock time, peak GPU memory, and wall-clock inference time per pair. MASt3R checkpoint is pretrained, so training time is not comparable (–). For deep ensembles, training time is summed over $K$ independently trained models.

| Method | Training (h) | PeakMem (GB) | Inference (ms/pair) |
|---|---|---|---|
| MCD ($T$=16) | – | 3.15 | 1225.77 |
| DeepEns ($K$=5) | 100.2 | 13.60 | 316.88 |
| MASt3R | – | 2.71 | 49.35 |
| Hetero | 9.4 | 2.88 | 70.14 |
| Trust3R | 10.5 | 3.16 | 80.92 |

between predicted and ground-truth pointmaps before computing geometry-based errors, so MAE/RMSE and ranking-based UQ metrics (AURC (Geifman et al., 2018)/AUSE (Ilg et al., 2018)/Spearman $\rho$ (Spearman, 1961)) reflect local geometric reliability rather than global pose/scale drift. All methods are evaluated on the same validity mask defined by available ground-truth depth/pointmap. We compute metrics per image and report dataset averages (NLL, when reported, is computed without alignment).

**Datasets.** We evaluate on diverse benchmarks chosen to stress-test uncertainty estimation across different capture conditions, spanning both indoor and outdoor scenes. Specifically, we use ScanNet++ (Yeshwanth et al., 2023) and TUM RGB-D (Sturm et al., 2012) for cluttered indoor RGB-D sequences with frequent occlusions and texture-poor regions, and KITTI (Geiger et al., 2013) and ETH3D (ablation mainly) (Schops et al., 2019) for outdoor/large-scale imagery with larger depth ranges, motion, and greater variability in scene structure and imaging conditions.

**Compared methods.** We compare against several single-pass baselines: **MASt3R** (Leroy et al., 2024), which uses the model's predicted confidence score as a heuristic uncertainty signal; and heteroscedastic Gaussian (**Hetero**, Lázaro-Gredilla & Titsias, 2011), which predicts per-pixel variance in a single forward pass. We also compare against sampling-based references: MC Dropout (**MCD**, Gal & Ghahramani, 2016) with $T$=16 stochastic forward passes, and Deep Ensembles (**DeepEns**, Lakshminarayanan et al., 2017) with $K$=5 independently trained models. Each ensemble member is trained with the same setup as the corresponding single-model baseline. Sampling-based UQ methods can provide strong uncertainty estimates, but their inference cost scales with $T$ or $K$, making them substantially more expensive for dense pointmap prediction.

**Metrics.** We adopted three commonly used metrics for UQ evaluation: area under the risk-coverage curve (AURC), area under the sparsification error curve (AUSE), and Spearman's rank correlation coefficient (Spearman $\rho$). The definitions are provided in Appendix D. Reconstruction accuracy is also reported with MAE/RMSE over valid pixels using Sim(3)-aligned 3D errors.

**Implementation details.** To ensure a fair comparison, all uncertainty baselines share the same MASt3R backbone, training data, and optimization schedule unless otherwise noted. We freeze the backbone and train only lightweight add-on heads. Optimization uses AdamW (weight decay 0.05) (Loshchilov & Hutter, 2017) with base learning rate $3 \times 10^{-4}$ (scaled by total batch size), with a cosine schedule over 10 epochs and no warmup. For evidential learning, we set evidence regularization $\lambda_{\text{evi}} = 10^{-3}$. Details can be found in Appendix B.

### 4.2. Uncertainty-aware Pointmap Prediction Results

**Uncertainty evaluation.** Table 1 reports uncertainty ranking performance across benchmarks. Figure 3 visualizes the corresponding risk–coverage and sparsification curves on three test sets, where Trust3R consistently yields lower ranking errors across a wide range of coverages. Among single-pass methods, Trust3R consistently achieves stronger

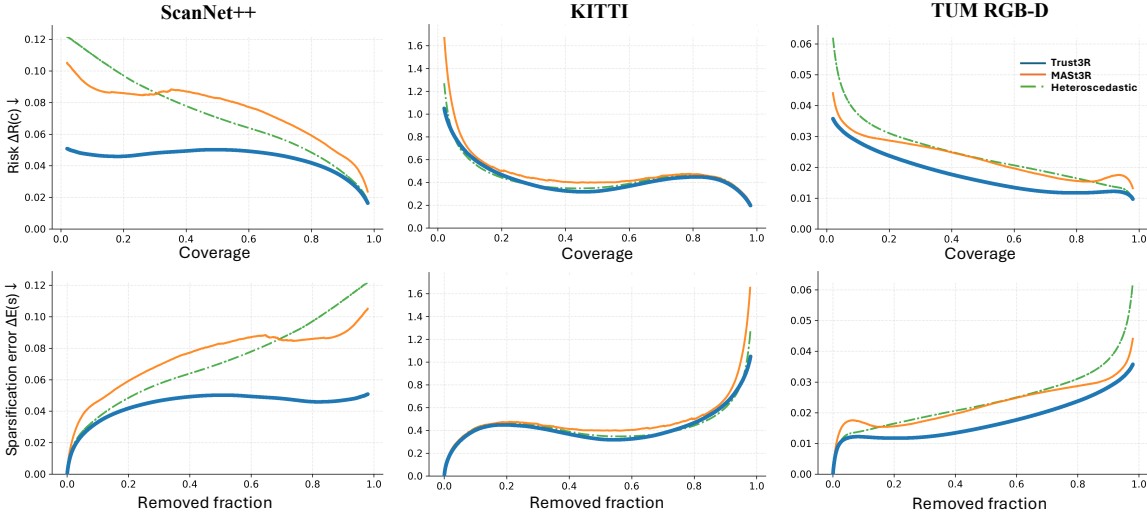

**Figure 3. Uncertainty ranking quality on three test sets.** Top row: *Risk–Coverage* $\Delta R(c) = R_{unc}(c) - R_{oracle}(c)$, where $R_{oracle}$ is obtained by sorting pixels by their *true* 3D error for the same method (method-specific oracle). Bottom row: *Sparsification error* $\Delta E(s) = E_{unc}(s) - E_{oracle}(s)$. Lower is better, $\Delta = 0$ indicates perfect (oracle) ranking. Each subplot uses its own y-axis range for readability.

selective ranking of high-error pixels, reflected by lower AURC and AUSE together with higher rank correlation. On ScanNet++, for instance, Trust3R significantly improves all three metrics, reducing AURC from 0.1649 to 0.1223 and AUSE from 0.0747 to 0.0444, while increasing rank correlation $\rho$ from 0.2837 to 0.4946. Figure 4 further illustrates typical cases, showing that Trust3R's predictive uncertainty aligns more closely with high-error regions than heuristic confidence scores in MASt3R.

**Reconstruction accuracy.** Apart from stronger uncertainty ranking quality in Table 1, Trust3R also improves MAE/RMSE of geometric accuracy on ScanNet++ and TUM RGB-D, as shown in Table 2. On KITTI, mean refinement can slightly increase MAE/RMSE relative to the frozen-mean baseline, reflecting a mild geometry–reliability trade-off, while uncertainty ranking remains improved and still prioritizes unreliable pixels even when the mean is imperfect.

**Efficiency.** Table 3 compares compute overhead. Trust3R remains single-pass (one model, one forward), matching the deployment cost of other single-pass baselines. In contrast, MC Dropout requires $T\times$ forwards and Deep Ensembles require $K\times$ models and forwards, making them substantially more expensive at inference.

**Ensemble inference comparison.** Deep Ensembles typically yield the strongest uncertainty metrics, but require training and running multiple models. Trust3R closes a substantial portion of this gap while remaining single-pass, which is important for downstream pipelines that cannot afford $K\times$ or $T\times$ inference. This makes Trust3R a practical drop-in uncertainty signal when downstream modules

*Table 4.* **Generalization to VGGT.** Higher is better for $\rho$; lower is better for AURC, AUSE, and NLL.

| Variant | $\rho \uparrow$ | AURC$\downarrow$ | AUSE$\downarrow$ | NLL$\downarrow$ |
|---|---|---|---|---|
| VGGT confidence | 0.3162 | 0.1084 | 0.0452 | -2.4770 |
| VGGT + Trust3R | **0.6419** | **0.0841** | **0.0209** | **-3.5999** |

require aggressive pruning under a fixed compute budget.

### 4.3. Cross-Architecture Validation on VGGT

To evaluate the cross-architecture generalization of our method, we attach the same lightweight uncertainty head to a frozen VGGT backbone and keep the training protocol unchanged. We compare against the original VGGT confidence signal using the same uncertainty-ranking and probabilistic scoring metrics. As shown in Table 4, adding the proposed head improves both uncertainty ranking and NLL on VGGT. These results suggest that the proposed evidential head can transfer beyond the MASt3R backbone in this setting.

### 4.4. Ablations

Our ablation study is designed to isolate which parts of Trust3R make the uncertainty useful for selective pointmap prediction. We study three choices: the scalar uncertainty readout, the evidential variational family, and the gated residual refinement of the predictive mean.

**Uncertainty readout.** Table 5 compares aleatoric uncertainty $u_{alea}$, total uncertainty $u_{total}$, and epistemic uncertainty $u_{epi}$ on ETH3D. All three scores are computed from

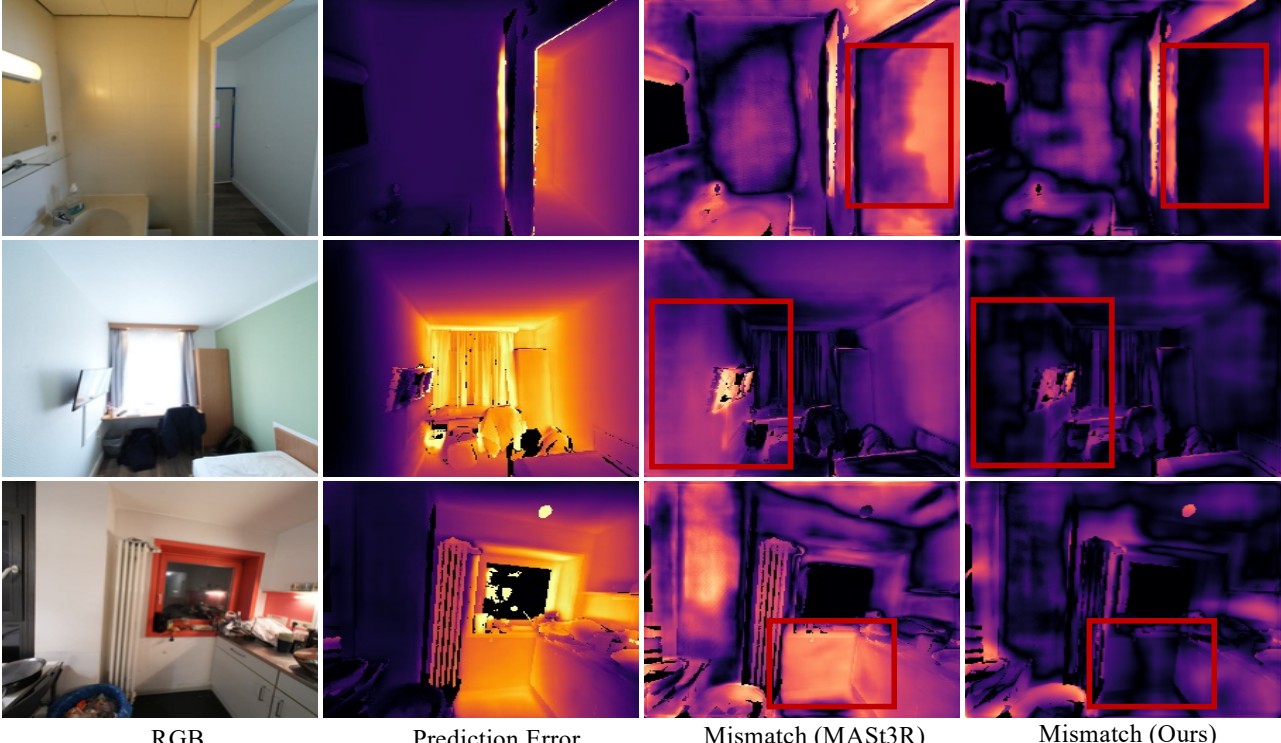

| RGB | Prediction Error | Mismatch (MASt3R) | Mismatch (Ours) |

*Figure 4.* **Qualitative uncertainty comparison.** (a) RGB input; (b) GT 3D point error; (c) mismatch map for MASt3R confidence; (d) mismatch map for Trust3R predictive uncertainty. Mismatch is defined as $d(i) = |p_u(i) - p_e(i)|$, where $p_u$ and $p_e$ are percentile ranks of uncertainty and GT error over valid pixels (lower is better). Darker pixels indicate lower mismatch (better uncertainty–error alignment), meaning unreliable high-error regions are more likely to be assigned high uncertainty and thus removed by uncertainty-based filtering.

the same predictive distribution, so the comparison only changes how the distribution is converted into a scalar ranking score. Aleatoric uncertainty mainly reflects local observation noise or ambiguity. It can be useful in difficult image regions, but it does not always indicate that the model has weak evidence for the 3D point. Total uncertainty improves over aleatoric uncertainty because it also includes the uncertainty of the predictive mean. Epistemic uncertainty gives the best ranking in this ablation, with the lowest AURC and AUSE and the highest Spearman correlation. We therefore use $u_{\text{epi}}$ as the default ranking measure in the remaining experiments.

**Evidential family.** Table 6 compares the factorized NIG head and the full-covariance NIW head. NIG treats the three point coordinates independently, while NIW allows the uncertainty of the $x$, $y$, and $z$ coordinates to be correlated. The difference is small on ScanNet++, where the indoor scenes are closer to the training distribution, but NIW still slightly improves all ranking metrics. The gain is clearer on ETH3D: NIW reduces AURC from 0.3213 to 0.3040 and AUSE from 0.1493 to 0.1318, while increasing Spearman correlation from 0.3229 to 0.3483. This suggests that cross-coordinate covariance is more useful when the test data is

*Table 5.* **Uncertainty measure ablation on ETH3D.** We compare aleatoric $u_{\text{alea}}$, epistemic $u_{\text{epi}}$, and total $u_{\text{total}}$ measures derived from the same predictive distribution.

| Readout | AURC↓ | AUSE↓ | $\rho$ ↑ |
|---|---|---|---|
| $u_{\text{alea}}$ | 0.3175 | 0.1452 | 0.3093 |
| $u_{\text{total}}$ | 0.3064 | 0.1341 | 0.3455 |
| $u_{\textbf{epi}}$ | **0.3040** | **0.1318** | **0.3483** |

*Table 6.* **Ablation on evidential variational family.** NIW models cross-axis correlations (full covariance), while NIG assumes independent coordinates (diagonal covariance).

| Test set | Variant | AURC↓ | AUSE↓ | $\rho$ ↑ |
|---|---|---|---|---|
| ScanNet++ | Trust3R (NIG) | 0.1237 | 0.0448 | 0.4875 |
| | Trust3R (NIW) | **0.1233** | **0.0444** | **0.4930** |
| ETH3D | Trust3R (NIG) | 0.3213 | 0.1493 | 0.3229 |
| | Trust3R (NIW) | **0.3040** | **0.1318** | **0.3483** |

more challenging or less aligned with the training domain. We use NIW as the default head because it provides a richer uncertainty representation while keeping inference single-pass.

**Gated residual refinement.** Table 7 shows that gated residual refinement improves uncertainty ranking on all

*Table 7.* **Effect of gated residual refinement.** We ablate whether the gated residual refinement is used for the predictive mean. Rows marked by ⋆ use gated residual refinement; unmarked rows use the frozen backbone mean.

| Dataset | GR | MAE↓ | RMSE↓ | AURC↓ | AUSE↓ |
|---------|----|------|-------|-------|-------|
| ScanNet++ | – | 0.2164 | 0.3026 | 0.1788 | 0.0887 |
|           | ⋆ | **0.2046** | **0.2926** | **0.1349** | **0.0512** |
| KITTI | – | **1.6108** | **3.0427** | 1.1170 | 0.6072 |
|       | ⋆ | 1.6436 | 3.0625 | **0.9942** | **0.4645** |
| ETH3D | – | 0.5607 | 0.8147 | 0.3453 | 0.1502 |
|       | ⋆ | **0.5366** | **0.7930** | **0.3347** | **0.1482** |
| TUM | – | 0.0938 | 0.1600 | 0.0576 | 0.0271 |
|     | ⋆ | **0.0909** | **0.1573** | **0.0496** | **0.0206** |

datasets and generally improves geometry. This controlled ablation setting is intended for within-table comparison rather than to duplicate the selected-model numbers in Table 2. KITTI is the only exception in geometric accuracy: MAE/RMSE slightly increase, while AURC and AUSE still improve, suggesting that in this out-of-domain outdoor setting, the refined mean may not always reduce reconstruction error, but the predicted uncertainty remains effective for ranking unreliable points.

### 4.5. Downstream Evaluation

Beyond intrinsic uncertainty-ranking metrics, we further evaluate whether the predicted uncertainty can serve as an actionable reliability signal in existing 3D pipelines. We test this in two settings: per-point weighting for MASt3R-SLAM pose estimation on TUM RGB-D, and reliability scoring for transparent and reflective regions on Tricky24.

**Uncertainty-weighted SLAM.** We integrate Trust3R uncertainty into the weighting path of MASt3R-SLAM on TUM RGB-D, while keeping the rest of the SLAM pipeline unchanged. This evaluates whether our uncertainty can improve a geometry optimization system that directly depends on reliable point correspondences. As shown in Table 8, replacing the original weighting signal with Trust3R uncertainty reduces ATE from 0.0287 to 0.0268 and RPE from 0.0321 to 0.0278. This result indicates that the predicted uncertainty can provide a useful reliability weight for downstream camera pose estimation.

**Transparent-scene reliability.** We next evaluate whether the uncertainty can identify difficult geometry in scenes with transparent and reflective objects. These materials often violate standard Lambertian assumptions and can cause severe depth or pointmap errors, especially near object boundaries (Zama Ramirez et al., 2024). We use

*Table 8.* **Downstream SLAM on TUM RGB-D.** We replace the original weighting signal in MASt3R-SLAM with Trust3R uncertainty and keep the remaining pipeline unchanged.

| Method | ATE↓ | RPE↓ |
|--------|------|------|
| MASt3R-SLAM baseline | 0.0287 | 0.0321 |
| + Trust3R uncertainty | **0.0268** | **0.0278** |

*Table 9.* **Transparent-scene reliability on Tricky24 (ring).** Mean over 32 scenes. Higher AUROC and lower FPR@95%TPR are better. Additional visualizations are provided in Appendix G.

| Trust signal | AUROC↑ | FPR@95%TPR↓ |
|--------------|--------|-------------|
| MASt3R | 0.4523 | 0.9146 |
| Trust3R | **0.4786** | **0.8788** |

Tricky24 and compute mask-based reliability metrics, AU-ROC and FPR@95%TPR, on a boundary ring band with radius $r=3$ pixels (details are provided in the appendix). We treat MASt3R confidence and our NIW uncertainty as reliability scores for detecting annotated transparent regions. As shown in Table 9, Trust3R achieves higher AUROC and lower FPR@95%TPR over 32 scenes, showing better awareness of unreliable geometry near refraction-induced boundaries.

## 5. Discussion and Limitations

Trust3R shows that per-point evidential uncertainty can be made useful for feed-forward pointmap reconstruction, but it is not a complete model of all geometric ambiguity. The current predictive distribution is unimodal at each 3D point. It therefore cannot explicitly represent multiple plausible 3D explanations in cases such as repeated patterns, strong reflections, or large occlusions. In these regions, the model can still mark points as unreliable, but it does not separate different hypotheses.

## 6. Conclusion

We presented Trust3R, a trust-aware 3D reconstruction framework that equips dense feed-forward pointmap prediction with accurate and probabilistically grounded uncertainty estimates. By combining evidential learning with multi-view geometry prediction, Trust3R yields a closed-form Student-$t$ predictive distribution whose uncertainty aligns closely with empirical geometric error, enabling reliable selective pointmaps. Across benchmarks, Trust3R consistently improves risk–coverage and sparsification behavior among single-pass methods without sacrificing geometric accuracy. Finally, our evidential uncertainty is used as an actionable reliability score for out-of-distribution awareness and failure-case identification, improving robustness in downstream reconstruction rather than merely producing qualitative uncertainty maps.

## Impact Statement

This work aims to improve the reliability of feed-forward 3D reconstruction by estimating where predicted geometry may be uncertain. More reliable uncertainty estimates can help downstream systems avoid over-trusting incorrect geometry, especially in applications such as mapping, visual SLAM, and robotic perception. At the same time, the proposed uncertainty should not be treated as a guarantee of correctness. The model may still fail under severe ambiguity, domain shift, occlusion, reflective or transparent materials, and other challenging imaging conditions. In safety-critical deployments, the uncertainty output should therefore be used together with additional sensor checks, task-level validation, and human oversight when appropriate. We do not identify additional societal impacts beyond those commonly associated with machine learning and 3D perception systems.

## Acknowledgements

The work of W. Zhao and C. Tian was supported in part by the National Science Foundation (NSF) via grants DMS-2312173 and ECCS-2433631. We also thank the anonymous reviewers and the area chair for their constructive feedback. Portions of this research were conducted with the advanced computing resources provided by Texas A&M High Performance Research Computing.

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

# A. Additional Method Details

Given an input image pair $(I_1, I_2)$, the frozen feed-forward geometry backbone predicts dense per-pixel 3D pointmaps $\hat{\mathbf{X}}_0^v \in \mathbb{R}^{H \times W \times 3}$ for view $v \in \{1, 2\}$, together with the original confidence map when available. Our uncertainty heads output either factorized evidential parameters (NIG) or multivariate evidential parameters (NIW). We denote a per-pixel 3D target as $\mathbf{x} \in \mathbb{R}^3$ and a scalar target as $y \in \mathbb{R}$.

**Scope and relation to prior evidential regression.** Our multivariate evidential formulation builds on the Normal–Inverse–Wishart evidential regression framework of Meinert & Lavin (2021). We do not claim the NIW prior itself as a new probabilistic model. Instead, our contribution is the pointmap-specific integration needed to make evidential uncertainty useful for dense feed-forward 3D reconstruction: a lightweight multivariate evidential head, gated residual mean refinement, stable evidence regularization, and local post-upsampling smoothing for token-to-pixel dense prediction.

**Trust-aware prediction.** Throughout the paper, "trust-aware" means that the model outputs a reliability signal that can be used to rank, filter, or weight predicted 3D points. It is not a binary guarantee that a point is correct. In our experiments, this reliability is evaluated through uncertainty–error ranking metrics, selective prediction curves, and downstream weighting.

**No repeated main-table results.** To avoid duplicating the main paper, this appendix does not repeat quantitative tables already reported in the main text, including the VGGT validation, gated residual ablation, evidential-family ablation, ETH3D uncertainty-readout ablation, MASt3R-SLAM, Tricky24, and the main compute table. Appendix tables below only provide additional rows, additional metrics, or additional protocols not already tabulated in the main paper.

**Uncertainty readout.** Many metrics require a scalar uncertainty score $u_i$ per pixel, with larger values indicating lower reliability. Unless stated otherwise, we use

$$u_{\text{conf},i} = -\log(\text{conf}_i + \epsilon) \tag{16}$$

for confidence-only baselines, and reduce XYZ uncertainty by trace,

$$u_i = \text{tr}(\Sigma_i) = \sum_{c \in \{x,y,z\}} \text{Var}[x_{i,c}], \tag{17}$$

for Gaussian, ensemble, NIG, and NIW variants. For the main NIW results and ablations, we use the epistemic trace unless a table explicitly states another readout.

## A.1. Multivariate Evidential Modeling via Normal–Inverse–Wishart

To model cross-coordinate correlations within each 3D point, we use a per-pixel Normal–Inverse–Wishart (NIW) evidential head:

$$\mathbf{m} \in \mathbb{R}^3, \qquad \kappa > 0, \qquad \nu > d + 1 \ (d = 3), \qquad \Psi \in \mathbb{R}^{3 \times 3}, \quad \Psi \succ 0.$$

The NIW prior is

$$\boldsymbol{\mu} \mid \boldsymbol{\Sigma} \sim \mathcal{N}(\mathbf{m}, \boldsymbol{\Sigma}/\kappa), \tag{18}$$

$$\boldsymbol{\Sigma} \sim \mathcal{W}^{-1}(\Psi, \nu). \tag{19}$$

After marginalizing $(\boldsymbol{\mu}, \boldsymbol{\Sigma})$, the predictive distribution is a multivariate Student-$t$:

$$p(\mathbf{x} \mid \mathbf{m}, \kappa, \nu, \Psi) = \text{St}_d(\mathbf{x}; \mathbf{m}, \Sigma_t, \nu_t),$$
$$\nu_t = \nu - d + 1, \tag{20}$$
$$\Sigma_t = \frac{\kappa + 1}{\kappa \nu_t} \Psi.$$

For $d = 3$, this gives $\nu_t = \nu - 2$, matching the main-text form in Eq. (7). Here $\Sigma_t$ is the Student-$t$ scale matrix. The predictive covariance exists for $\nu_t > 2$ and is

$$\text{Cov}[\mathbf{x}] = \frac{\nu_t}{\nu_t - 2} \Sigma_t = \frac{\kappa + 1}{\kappa(\nu - d - 1)} \Psi. \tag{21}$$

**Positive-definite parameterization.** We parameterize the scale matrix by a Cholesky factor:

$$\Psi = \mathbf{L}\mathbf{L}^\top. \tag{22}$$

The head regresses the six lower-triangular entries of $\mathbf{L}$ per pixel. Diagonal entries are mapped through $\text{softplus}(\cdot) + \epsilon$ to ensure positivity, while off-diagonal entries are unconstrained. We also enforce

$$\kappa = \text{softplus}(\cdot) + \epsilon, \qquad \nu = (d+1) + \text{softplus}(\cdot),$$

so that $\nu > d + 1$.

**Multivariate Student-$t$ negative log-likelihood.** Let

$$\delta = (\mathbf{x} - \mathbf{m})^\top \Sigma_t^{-1} (\mathbf{x} - \mathbf{m}).$$

The log-density is

$$\log p(\mathbf{x}) = \log \Gamma\left(\frac{\nu_t + d}{2}\right) - \log \Gamma\left(\frac{\nu_t}{2}\right) - \frac{1}{2}\left(d \log(\nu_t \pi) + \log |\Sigma_t|\right)$$
$$- \frac{\nu_t + d}{2} \log\left(1 + \frac{\delta}{\nu_t}\right). \tag{23}$$

We minimize $\mathcal{L}_{\text{NLL}} = -\log p(\mathbf{x})$ over valid pixels.

**NIW evidential regularizer.** Following evidential learning, we penalize high evidence when the predictive mean is wrong:

$$\mathcal{L}_{\text{evi}} = \|\mathbf{x} - \mathbf{m}\|_2^2 (\kappa + \nu). \tag{24}$$

The NIW loss is

$$\mathcal{L}_{\text{NIW}} = \mathcal{L}_{\text{NLL}} + \lambda_{\text{evi}} \mathcal{L}_{\text{evi}}. \tag{25}$$

**Aleatoric and epistemic decomposition.** For $\nu > d + 1$, NIW gives the closed-form decomposition

$$\Sigma_{\text{alea}} = \mathbb{E}[\Sigma] = \frac{\Psi}{\nu - d - 1}, \tag{26}$$

$$\Sigma_{\text{epi}} = \text{Var}[\boldsymbol{\mu}] = \frac{\Psi}{\kappa(\nu - d - 1)}, \tag{27}$$

$$\Sigma_{\text{tot}} = \Sigma_{\text{alea}} + \Sigma_{\text{epi}} = \frac{\kappa + 1}{\kappa(\nu - d - 1)} \Psi. \tag{28}$$

For scalar ranking, we use the trace of these matrices.

## A.2. Residual-Gated Mean Refinement

The evidential likelihood depends on the predictive mean. Directly fine-tuning a large pretrained pointmap backbone with an evidential objective may degrade the pretrained geometry and can produce unstable evidence. We therefore use a lightweight residual-gated refinement branch:

$$\hat{\mathbf{X}}_{\text{ref}}^v = \hat{\mathbf{X}}_0^v + \sigma(\mathbf{g}^v) \odot \Delta^v, \tag{29}$$

where $\mathbf{g}^v$ is a gate-logit map, $\Delta^v$ is a 3D residual, and $\sigma(\cdot)$ is the sigmoid function. Both $\mathbf{g}^v$ and $\Delta^v$ are predicted from the concatenated encoder/decoder tokens using lightweight heads and reshaped to the pixel grid. We initialize the residual branch near zero so that the model starts from the pretrained pointmap prediction.

The purpose of this module is not parameter-efficient fine-tuning in the usual PEFT sense. Its role is to selectively refine the predictive mean only where evidential training needs it, while leaving reliable pretrained geometry mostly unchanged.

### A.3. Post-upsampling Smoothing

Dense heads that upsample ViT tokens to pixels can introduce patch-grid artifacts in the residual branch. To reduce this local artifact, we optionally apply a lightweight post-upsampling smoothing operator:

$$\Delta \tilde{\mathbf{m}}^v = S_\Delta(\mathrm{Up}(\Delta_{\mathrm{tok}}^v)), \tag{30}$$

$$\tilde{\mathbf{g}}^v = S_g(\mathrm{Up}(\mathbf{g}_{\mathrm{tok}}^v)), \tag{31}$$

$$\hat{\mathbf{X}}_{\mathrm{ref}}^v = \hat{\mathbf{X}}_0^v + \sigma(\tilde{\mathbf{g}}^v) \odot \Delta \tilde{\mathbf{m}}^v. \tag{32}$$

Here $S_\Delta$ and $S_g$ are depthwise-separable $3\times3$ convolutions, optionally followed by a $1\times1$ convolution. They are initialized close to identity, so the refinement branch remains a near no-op at the start of training. This smoothing is applied to the residual mean-refinement path, not directly to the evidential parameters. Thus, it introduces only local regularization of the refined mean and does not model explicit cross-pixel covariance.

### A.4. Factorized Evidential Modeling via Normal–Inverse–Gamma

For scalar regression, we use Normal–Inverse–Gamma (NIG) evidence:

$$\gamma \in \mathbb{R}, \qquad \nu > 0, \qquad \alpha > 1, \qquad \beta > 0.$$

The induced Student-$t$ predictive distribution is

$$p(y \mid \gamma, \nu, \alpha, \beta) = \mathrm{St}\left(y; \mu = \gamma, \lambda = \frac{\beta(1+\nu)}{\nu\alpha}, \mathrm{df} = 2\alpha\right), \tag{33}$$

with predictive mean and variance

$$\mathbb{E}[y] = \gamma, \qquad \mathrm{Var}[y] = \frac{\beta(1+\nu)}{\nu(\alpha-1)}. \tag{34}$$

**Scalar NIG NLL.** The negative log-likelihood is

$$\begin{aligned}
\mathcal{L}_{\mathrm{NLL}} = &\frac{1}{2}\log\left(\frac{\pi}{\nu}\right) - \alpha\log(2\beta(1+\nu)) \\
&+ \left(\alpha + \frac{1}{2}\right)\log\left(\nu(y-\gamma)^2 + 2\beta(1+\nu)\right) + \log\Gamma(\alpha) - \log\Gamma\left(\alpha + \frac{1}{2}\right).
\end{aligned} \tag{35}$$

**Scalar evidence regularizer.**

$$\mathcal{L}_{\mathrm{evi}} = |y - \gamma|(2\nu + \alpha), \qquad \mathcal{L}_{\mathrm{NIG}} = \mathcal{L}_{\mathrm{NLL}} + \lambda_{\mathrm{evi}}\mathcal{L}_{\mathrm{evi}}. \tag{36}$$

**Factorized XYZ-NIG.** For dense 3D points $\mathbf{x} = (x, y, z)$, XYZ-NIG assumes conditional independence across coordinates:

$$p(\mathbf{x} \mid \Theta) = \prod_{c \in \{x,y,z\}} p(x_c \mid \gamma_c, \nu_c, \alpha_c, \beta_c), \tag{37}$$

and minimizes the coordinate-averaged NIG loss:

$$\mathcal{L}_{\mathrm{XYZ\text{-}NIG}} = \frac{1}{3} \sum_{c \in \{x,y,z\}} \mathcal{L}_{\mathrm{NIG}}(x_c; \gamma_c, \nu_c, \alpha_c, \beta_c). \tag{38}$$

For each coordinate, a convenient decomposition is

$$\sigma_{\mathrm{alea}}^2 = \frac{\beta}{\alpha-1}, \qquad \sigma_{\mathrm{epi}}^2 = \frac{\beta}{\nu(\alpha-1)}, \qquad \sigma_{\mathrm{tot}}^2 = \frac{\beta(1+\nu)}{\nu(\alpha-1)}. \tag{39}$$

For XYZ ranking, we sum the corresponding coordinate variances.

# B. Implementation Details

**Backbone and training protocol.** Unless stated otherwise, all uncertainty variants use the same MASt3R backbone, training data, and optimization schedule. We freeze the pretrained backbone and geometry heads and train only lightweight uncertainty/refinement heads. This design follows the goal of equipping a strong pretrained pointmap predictor with uncertainty rather than relearning geometry from scratch.

**Optimization.** We use AdamW with weight decay $0.05$ and base learning rate $3 \times 10^{-4}$, scaled by total batch size. Training uses a cosine learning-rate schedule for 10 epochs with no warmup. The effective batch size is 10. For evidential learning, we set

$$\lambda_{\mathrm{uq}}^{xyz} = 0.05, \qquad \lambda_{\mathrm{evi}}^{xyz} = 10^{-3}.$$

The loss is summed over both views and averaged over valid pixels:

$$\mathcal{L} = \lambda_{\mathrm{uq}} \left( \mathcal{L}_{\mathrm{NLL}} + \lambda_{\mathrm{evi}} \mathcal{L}_{\mathrm{evi}} \right). \tag{40}$$

**Training data.** We train on the mix4 data mixture: 25k ScanNet++ pairs, 25k ARKitScenes pairs, 50k Waymo pairs, and 50k MegaDepth pairs. The input resolution is 224 with crop augmentation parameter `aug_crop`= 16. Here, `aug_crop` denotes the random crop margin used during training augmentation. We validate on 2k ScanNet++ pairs with `aug_crop`= 0.

**MC Dropout and Deep Ensemble settings.** MC Dropout uses $T = 16$ stochastic forward passes. Deep Ensemble uses $K = 5$ independently trained models. Each ensemble member uses the same training script and setup as the corresponding baseline, with independent training runs. Larger $T$ or $K$ may improve sampling-based estimates but directly increases inference cost.

**Fairness of single-pass baselines.** Single-pass baselines use the same backbone and data. Heteroscedastic regression predicts a per-pixel Gaussian variance in one forward pass. XYZ-NIG and XYZ-NIW differ in the evidential parameterization: XYZ-NIG assumes independent coordinates, while XYZ-NIW models a full per-point covariance through the NIW prior. Unless explicitly marked as a frozen-mean baseline, variants use the same residual-gated mean-refinement design and optimization schedule.

**Ensemble memory/latency interpretation.** Deep Ensembles incur either higher latency or higher peak memory depending on implementation. Running $K$ models sequentially keeps memory closer to one model but increases latency by roughly $K$ forward passes. Running them in parallel can reduce wall-clock latency but requires approximately $K$ model copies in memory. The main compute table follows a fixed implementation protocol, while the key comparison is that Trust3R remains a single-model, single-forward method.

# C. Datasets and Preprocessing

We evaluate on held-out scenes and cross-dataset test sets. To avoid ambiguity, we distinguish held-out in-benchmark evaluation from cross-dataset evaluation.

**ScanNet++.** ScanNet++ is used both in training and evaluation, but train/validation/test scenes are disjoint. Thus, ScanNet++ measures generalization to unseen scenes within the same benchmark rather than strict out-of-domain generalization. We use preprocessed metadata containing scene id, image, depth, intrinsics, pose, and pair lists. RGB images and depth maps are resized to the network resolution, and intrinsics are adjusted accordingly. Pixels with missing or invalid depth are excluded from training and evaluation masks.

**TUM RGB-D, KITTI, and ETH3D.** TUM RGB-D, KITTI, and ETH3D are not used for the reported training mixture and are therefore cross-dataset evaluations. They differ from the training data in capture setup, depth range, camera motion, and scene statistics. We form image pairs using dataset-provided pair lists or fixed-stride temporal pairing and apply dataset validity masks.

# D. Evaluation Protocol and Metrics

We evaluate dense pointmap prediction over valid pixels. For pointmap accuracy and uncertainty-ranking metrics, we apply the same per-image Sim(3) alignment between predicted and ground-truth pointmaps as in the main paper. This makes MAE/RMSE, AURC, AUSE, and Spearman $\rho$ reflect local geometric reliability rather than global pose/scale drift. For probabilistic scoring, we compute NLL in the raw camera frame without post-hoc alignment.

**Notation.** Let $\Omega$ be the valid-pixel set and $N = |\Omega|$. For each pixel $i \in \Omega$, define the 3D error

$$e_i = \|\hat{\mathbf{x}}_i - \mathbf{x}_i^\star\|_2 , \tag{41}$$

and a scalar uncertainty score $u_i$, where larger $u_i$ means less reliable.

**Pointmap MAE/RMSE.**

$$\mathrm{MAE}_{3D} = \frac{1}{N} \sum_{i \in \Omega} e_i, \tag{42}$$

$$\mathrm{RMSE}_{3D} = \sqrt{\frac{1}{N} \sum_{i \in \Omega} e_i^2}. \tag{43}$$

**Spearman rank correlation.**

$$\rho = \mathrm{corr}\left(\mathrm{rank}(u), \mathrm{rank}(e)\right). \tag{44}$$

Higher $\rho$ indicates better monotonic alignment between predicted uncertainty and actual point error.

**Risk–coverage and AURC.** For coverage $c \in (0, 1]$, retain the $\lfloor cN \rfloor$ pixels with lowest uncertainty. The risk is

$$R(c) = \frac{1}{|\Omega_c|} \sum_{i \in \Omega_c} e_i. \tag{45}$$

AURC is the area under $R(c)$, computed by numerical integration on a uniform coverage grid. Lower is better.

**Sparsification and AUSE.** For sparsification fraction $s$, remove the top $\lceil sN \rceil$ pixels by uncertainty. Let $E_{\mathrm{unc}}(s)$ be the remaining mean error. The oracle removes pixels by true error and gives $E_{\mathrm{oracle}}(s)$. We report

$$\mathrm{AUSE} = \int_0^1 \left(E_{\mathrm{unc}}(s) - E_{\mathrm{oracle}}(s)\right) ds. \tag{46}$$

Lower is better, and zero corresponds to oracle ranking.

**Negative log-likelihood.** When a method defines a predictive density, we report

$$\mathrm{NLL} = -\frac{1}{N} \sum_{i \in \Omega} \log p_\theta(\mathbf{x}_i^\star). \tag{47}$$

Confidence-only baselines do not define a predictive likelihood and are omitted from NLL tables. For MC Dropout and Deep Ensembles, we moment-match sampled predictions to a diagonal Gaussian and add a small noise floor $\sigma_0^2$ selected on ScanNet++ validation. This noise floor is used only for NLL and does not affect ranking metrics.

**Point-cloud metrics.** In addition to per-pixel MAE/RMSE, we report point-cloud metrics on aligned dense pointmaps: Accuracy, Completeness, Chamfer Distance (CD), and F1 at threshold $0.05$. Accuracy is the mean nearest-neighbor distance from predicted points to ground-truth points. Completeness is the mean nearest-neighbor distance from ground-truth points to predicted points. CD is the average of Accuracy and Completeness. F1@0.05 is the harmonic mean of precision and recall under the $0.05$ distance threshold. We do not report Normal Consistency because the evaluated models predict dense 3D pointmaps rather than oriented surface normals; estimating normals introduces an additional design choice that is outside the pointmap protocol.

*Table 10.* **Additional reconstruction accuracy for UQ baselines not tabulated in the main accuracy table.** We report Sim(3)-aligned pointmap MAE/RMSE. MASt3R and Trust3R rows are in Main Table 2; Heteroscedastic shares the frozen MASt3R mean and is therefore not repeated.

| Dataset | Method | MAE↓ | RMSE↓ |
|---|---|---|---|
| ScanNet++ | MC Dropout | 0.2962 | 0.3809 |
| | Deep Ensemble | 0.1131 | 0.2095 |
| TUM RGB-D | MC Dropout | 0.1405 | 0.1986 |
| | Deep Ensemble | 0.0695 | 0.1220 |

# E. Downstream Evaluation Details

## E.1. Uncertainty-weighted MASt3R-SLAM

For MASt3R-SLAM, we replace the original weighting signal with Trust3R uncertainty while keeping the rest of the pipeline unchanged. The quantitative ATE/RPE results are reported in the main paper, so they are not repeated here. This appendix only clarifies the integration: the uncertainty map is converted into a per-point reliability weight and used in the same weighting path as the original confidence signal. No additional pose optimization terms or extra learned modules are introduced.

## E.2. Tricky24 Transparent-scene Reliability

We evaluate on 32 Tricky24 scenes with provided transparent-object masks. The main paper reports the quantitative AUROC and FPR@95%TPR table, so the numbers are not repeated here. For MASt3R confidence, we use $u = -\log(\text{conf} + \epsilon)$. For Trust3R, we use the trace of the predicted NIW uncertainty.

**Boundary ring construction.** To focus on boundary failures around transparent and reflective objects, we build a ring band of radius $r = 3$ pixels by morphology:

$$\text{ring} = \text{dilate}(m, r) \wedge \neg \text{erode}(m, r). \tag{48}$$

**Metrics.** Pixels in the ring are treated as positives and all remaining pixels as negatives. We compute AUROC and FPR@95%TPR per scene and report the mean over scenes in the main paper.

# F. Additional Results Not Duplicated in the Main Text

## F.1. Additional Reconstruction Accuracy for UQ Baselines

Main Table 2 reports the primary reconstruction accuracy comparison for MASt3R and Trust3R. To avoid repeating those rows, Table 10 only reports additional UQ baselines that are not tabulated in Main Table 2. The heteroscedastic baseline keeps the frozen MASt3R mean and only adds a variance head; therefore, its MAE/RMSE are identical to the MASt3R row in Main Table 2 and are not duplicated here.

Deep Ensembles can improve reconstruction accuracy because they aggregate multiple independently trained models, but this requires substantially more training and inference resources. Trust3R is designed primarily for single-pass uncertainty estimation and improves over the MASt3R mean prediction while keeping one model and one forward pass.

## F.2. Point-cloud Metrics

Table 11 reports point-cloud metrics on KITTI using the same aligned dense pointmaps. These metrics complement the per-pixel MAE/RMSE metrics with global point-cloud evaluation.

Trust3R achieves the best F1@0.05 and second-best CD/Accuracy/Completeness under this protocol, showing that its reliability improvement does not come at the cost of collapsed point-cloud quality.

*Table 11.* **Point-cloud metrics on KITTI.** F1 is computed at threshold 0.05. CD denotes Chamfer Distance. Best results are bolded and second-best results are underlined.

| Method | F1@0.05↑ | CD↓ | Accuracy↓ | Completeness↓ |
|---|---|---|---|---|
| MASt3R | 0.0627 | 0.5440 | 0.6371 | 0.4509 |
| Heteroscedastic | 0.0627 | 0.5440 | 0.6371 | 0.4509 |
| MC Dropout | 0.0345 | **0.4287** | **0.5090** | **0.3484** |
| Deep Ensemble | 0.0627 | 0.5443 | 0.6394 | 0.4492 |
| Trust3R | **0.0718** | 0.4904 | 0.5327 | 0.4481 |

*Table 12.* **Additional NIW uncertainty-source readouts.** This table reports additional total and aleatoric readouts on ScanNet++, TUM RGB-D, and KITTI. The selected epistemic readout is reported in the main uncertainty table and is not repeated here. Higher is better for $\rho$; lower is better otherwise.

| Dataset | Source | AURC↓ | AUSE↓ | $\rho$ ↑ |
|---|---|---|---|---|
| ScanNet++ | Total | 0.1253 | 0.0464 | 0.4725 |
| | Aleatoric | 0.1264 | 0.0475 | 0.4605 |
| TUM RGB-D | Total | 0.0491 | 0.0188 | 0.4986 |
| | Aleatoric | 0.0493 | 0.0190 | 0.4927 |
| KITTI | Total | 1.0111 | 0.4673 | 0.4481 |
| | Aleatoric | 1.1264 | 0.5826 | 0.3749 |

## F.3. Additional Uncertainty-source Results

Main Table 5 already reports aleatoric, total, and epistemic readouts on ETH3D. To avoid duplicating that table, Table 12 reports only the additional non-selected readouts on ScanNet++, TUM RGB-D, and KITTI. The selected epistemic readout is used in the main uncertainty results and is therefore not repeated here.

Together with the selected epistemic results in the main uncertainty table and the ETH3D decomposition in Main Table 5, these additional rows show that epistemic uncertainty is generally the most informative reliability signal under our ranking protocol.

## F.4. Inference and Calibration Diagnostics

Table 13 compares the inference-only overhead of NIG and NIW heads under the same profiling protocol. NIW adds a full per-point covariance parameterization but increases latency by only about 1 ms compared with NIG in this micro-benchmark.

*Table 13.* **Inference-only overhead of NIG vs. NIW heads.** This micro-benchmark isolates head overhead and is separate from the end-to-end timing in Main Table 3.

| Variant | Latency (ms) | Peak memory (MB) | Extra vs. NIG |
|---|---|---|---|
| XYZ-NIG | 54.37 | 6005.5 | – |
| XYZ-NIW | 55.49 | 6266.8 | +1.12 ms / +261.3 MB |

Table 14 reports a component-wise latency micro-benchmark. It isolates the added cost of the evidential head and gated residual branch. This table should be interpreted as a component-level profile rather than as a replacement for the end-to-end compute table in the main paper.

*Table 14.* **Component-wise single-pass latency.** All variants use one model and one forward pass.

| Variant | Latency (ms) | Overhead vs. MASt3R |
|---|---|---|
| MASt3R baseline | 52.6 | – |
| + Evidential UQ head | 57.5 | +4.9 ms (+9.3%) |
| + Evidential UQ + gated residual | 58.2 | +5.6 ms (+10.6%) |

Table 15 reports NLL for probabilistic baselines. NLL is evaluated in the raw camera frame without Sim(3) alignment, so it measures probabilistic calibration rather than aligned geometric error.

*Table 15.* **NLL across datasets.** Lower is better. Confidence-only baselines do not define a likelihood and are omitted.

| Method | ScanNet++ | TUM | KITTI | ETH3D |
|---|---|---|---|---|
| Heteroscedastic (Gauss) | -0.8374 | -4.2523 | 3.5102 | 37.4365 |
| XYZ-NIG | -0.6852 | -1.4672 | **1.1738** | **7.8879** |
| XYZ-NIW | **-3.0020** | **-5.1541** | 9.4236 | 11.6632 |

NIW has the best NLL on ScanNet++ and TUM, but its NLL is worse on KITTI and ETH3D. This does not imply a collapse of the aligned geometry or ranking metrics, because NLL is computed without alignment while MAE/RMSE and uncertainty-ranking metrics are computed after alignment. On KITTI, the degradation is concentrated at long range: pixels beyond 10 m account for 77.7% of valid pixels and 79.3% of NIW NLL. We therefore interpret the KITTI/ETH3D NLL behavior mainly as long-range calibration difficulty in outdoor scenes.

## G. Qualitative Results

**Transparent-scene uncertainty maps.** Figures 5–6 provide additional Tricky24 examples with uncertainty maps rather than only mismatch maps. Each figure shows RGB, oracle 3D error, and uncertainty maps, so readers can inspect both the scale and spatial structure of predicted uncertainty.

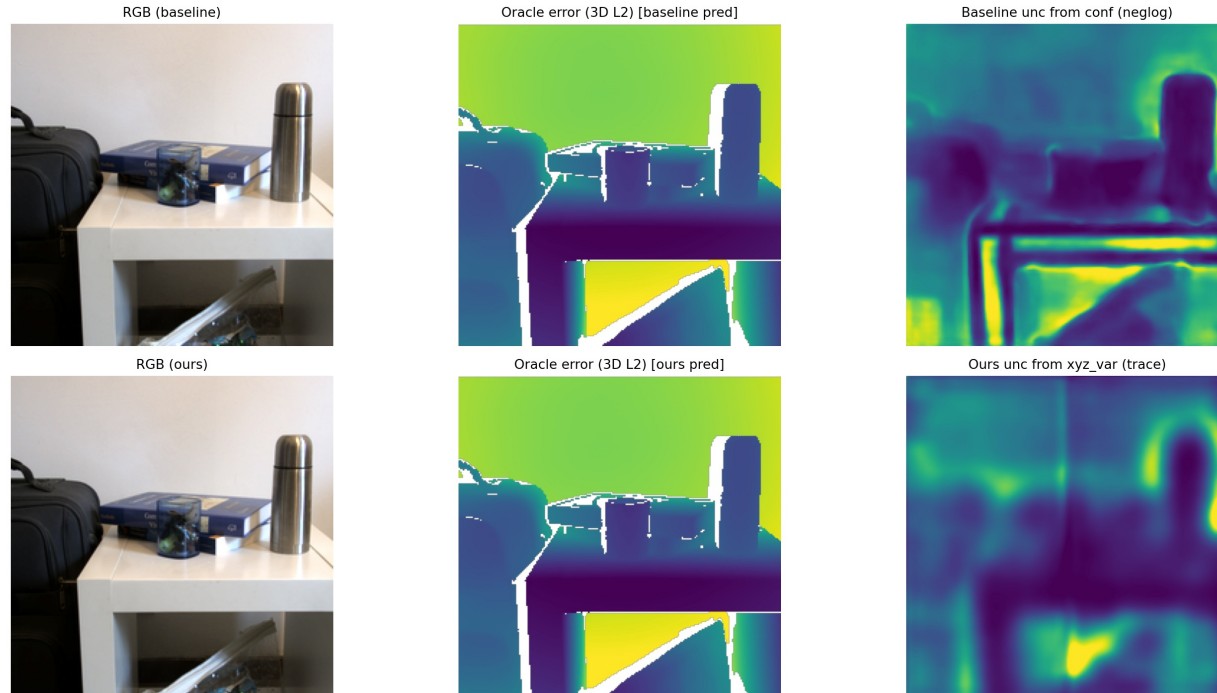

*Figure 5.* **Qualitative reliability on Tricky24, example A.** Top: MASt3R baseline; bottom: Trust3R. Columns show RGB, oracle 3D error, and the corresponding uncertainty map.

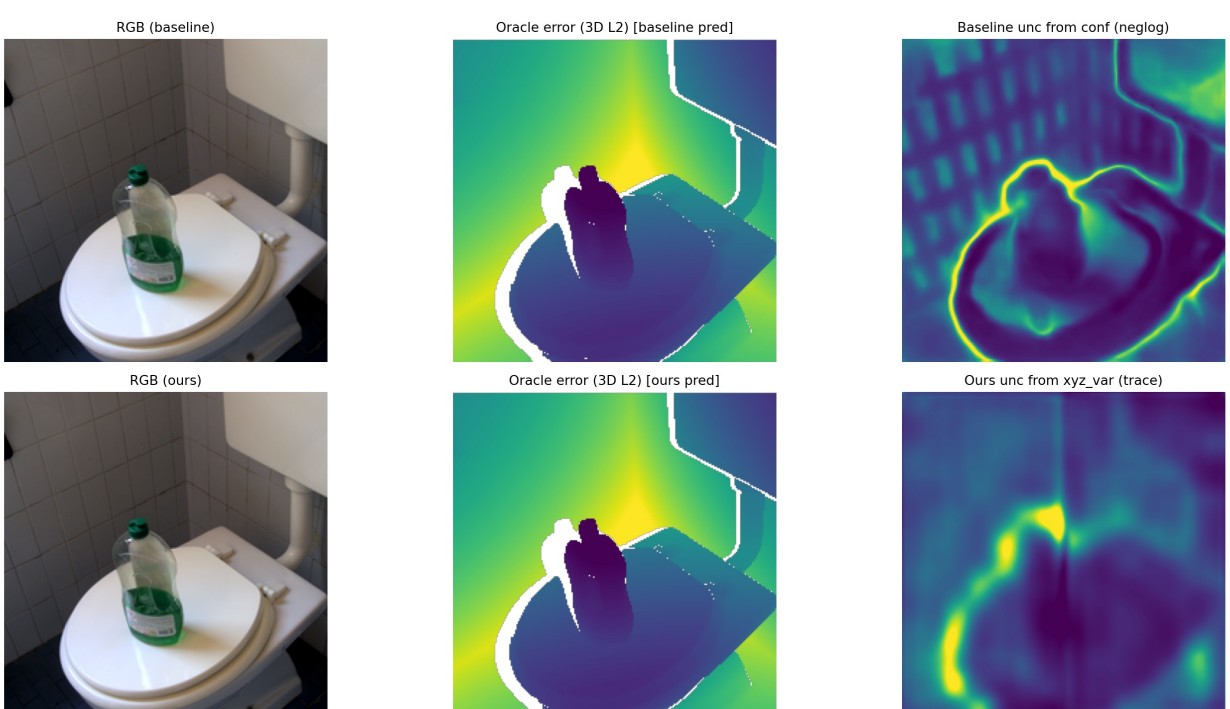

*Figure 6.* **Qualitative reliability on Tricky24, example B.** Top: MASt3R baseline; bottom: Trust3R. Columns show RGB, oracle 3D error, and the corresponding uncertainty map.

## H. Limitations and Practical Notes

Trust3R models each 3D point with a unimodal predictive distribution and a mean-field factorization across pixels. The NIW head models cross-coordinate covariance within each point, but it does not explicitly model dense cross-pixel covariance. Spatial structure is still captured implicitly by the transformer backbone and local residual smoothing, but explicit spatial covariance modeling is left for future work.

Evidential uncertainty can also become difficult to calibrate in long-range outdoor scenes, as reflected by the KITTI/ETH3D NLL results. In practice, we recommend using validation-set calibration checks, monitoring long-range bins separately, and treating uncertainty as a reliability weight rather than an absolute correctness guarantee.

## I. Reproducibility

We will release training and evaluation scripts, model checkpoints, dataset preprocessing instructions, and metric computation code. The release should include the cross-architecture VGGT experiment and the MASt3R-SLAM downstream weighting experiment, since these components are important for reproducing the rebuttal-added results.

