# OpenReview forum: "Trust It or Not: Evidential Uncertainty for Feed-Forward 3D Reconstruction with Trust3R"
_ICML.cc/2026/Conference — ICML 2026 regular_

### Official Review · Reviewer_siuW · 2026-02-23

**Soundness:** 3
**Presentation:** 3
**Significance:** 3
**Originality:** 3
**Overall Recommendation:** 4
**Confidence:** 4

**Summary:**

The paper introduces Trust3R, a framework that applies deep evidential regression to MASt3R model to quantify uncertainty of point cloud predictions. Each 3D point in the point cloud is assumed to follow a multivariate Gaussian distribution with mean and covariance following a Normal-Inverse-Wishart (NIW) prior. Through this formulation, the aleatoric and epistemic uncertainty of the point predictions can be quantified in closed-form from the predicted distributional parameters. Experimental results across indoor and outdoor benchmarks demonstrate that the approach provides more accurate uncertainty estimates than several baseline methods on MASt3R.

**Compliance With Llm Reviewing Policy:**

Affirmed.

**Final Justification:**

Considering the merits of this paper and the resolved concerns, I am willing to raise the score to weak accept. I strongly encourage the authors to add all the additional results mentioned in the rebuttal to the revision.

**Key Questions For Authors:**

1. For deep ensembles, how was each individual model trained? Was each model trained using the exact script that produced the MASt3R baseline?
2. Are the Herero and NIG baselines also trained with all the other components of the proposed NIW method, including gated residual head, post-smoothing, etc.? This is important to ensure a fair comparison.
3. See limitations.

**Limitations:**

Sec. 5 discusses some limitations of the method, e.g., the evidential model can be unstable in optimization and may have saturating uncertainty. I expect these issues to be further examined and analyzed, as they are important for application of the approach. When would these problems tend to occur and how to avoid them in practice? More visual examples would greatly help readers understand the failure modes.

**Strengths And Weaknesses:**

Strengths
1. The proposed method is well-motivated. Many current multi-view feedforward 3D reconstruction methods (MASt3R, VGGT, etc.) focuses only on reconstruction performance without caring much about the reliability of the predictions. These methods typically use unprincipled and heuristic confidence estimate that lacks a probabilistic interpretation and often aligns poorly with the actual predictive errors. This research direction is potentially of great significance to the 3D vision community.
2. The proposed method is mathematically sound. As the probabilistic framework itself has been established on general multivariate regression tasks, its application to point cloud uncertainty quantification is straightforward.
3. The experiments cover multiple baselines and datasets.

Weakness
1. The methodology relies heavily on the Multivariate Deep Evidential Regression framework. (Meinert, N. and Lavin, A., 2021. Multivariate deep evidential regression. arXiv preprint arXiv:2104.06135.) While I noticed that this paper is currently still on arXiv, given that this work has been public for several years and is the primary source for the multivariate extension of deep evidential regression, I still think that it should be at least cited and discussed to properly position Trust3R's technical contributions.
2. Some important ablation studies are missing.
- Making the entire model trainable vs. freezing the backbone with a residual head.
- Gated head for residual modulation.
- Effect of post-upsampling smoothing mentioned in A.3.
- The choice of K and T in MC dropout and deep ensembles.
3. Evaluation issues.
- Most qualitative figures only show the mismatch maps instead of uncertainty maps, hindering the evaluation of the scale of the uncertainty estimates.
- Tab. 1 and 2: Reconstruction performance should be analyzed for all the baselines in Tab. 1.
- Tab. 2: The evaluation metrics chosen (MAE/RMSE) are less standard for 3D point cloud assessment than metrics like Accuracy, Completeness, Normal Consistency, etc.
- Tab. 2: KITTI results missing. Sec. 4.2 states that "on KITTI, mean refinement can slightly increase MAE/RMSE relative to the frozen-mean baseline". The authors should not hide these inferior results from Tab. 2, but should clearly show the performance gap, in order for readers to better understand the method's failure mode.
- Tab. 3: The inference cost evaluation for Deep Ensembles is not fair. Compared to baseline MASt3R, if K models are running in parallel, peak memory should be K times higher but the inference time should be similar; If they are running in sequence, inference time can be K times longer but peak memory should be similar. But the result seems to show that both inference time and memory are much higher than MASt3R.
- Tab. 4: Aleatoric, epistemic, and total uncertainty's behavior on different scenarios is very important. Results on other datasets should be given in at least supp. mat.
- Tab 5: Computational overhead of NIG vs. NIW can be provided for evaluation of the performance-efficiency tradeoff.
4. The declared scope in the title and paper is too board. "Feed-forward pointmap prediction" currently includes far more methods (especially multi-view ones like VGGT) than just MASt3R. However, the experiments have MASt3R as the only backbone.
5. Some notations used might come from previous work and are not clearly defined in this paper, e.g., $H^v$ in Eq. 13, aug_crop in L684.

---

> ### Author Rebuttal · Authors · 2026-03-31
>
> We thank the reviewer for the feedback and address each concern below.
>
> **[W1] Positioning with prior multivariate evidential regression**
> We thank the reviewer for pointing this out and will cite Multivariate Deep Evidential Regression. Our claim is not that NIW itself is new, but that adapting evidential regression to dense feed-forward pointmap prediction requires pointmap-specific design. Our contribution is the integration recipe for this setting, including a lightweight evidential head, gated mean refinement, and a stable training design.
>
> **[W2] Missing ablation studies**
> - **Making the entire model trainable.** Our goal is not to relearn geometry, but to equip a strong pretrained pointmap model with probabilistically grounded uncertainty. Full retraining is unnecessary and may forget useful geometric priors, so we freeze the backbone and train lightweight add-on heads.
>
> - **Gated residual head.** The ablation below shows that gated residual refinement improves both geometry and uncertainty ranking on ScanNet++, ETH3D, and TUM. On KITTI, it causes a small MAE/RMSE trade-off but still improves uncertainty ranking.
>
> | Dataset | Gated Residual | MAE ↓ | RMSE ↓ | AURC ↓ | AUSE ↓ |
> |---|---|---:|---:|---:|---:|
> | ScanNet++ | w/o | 0.2164 | 0.3026 | 0.1788 | 0.0887 |
> |  | w/ | **0.2046** | **0.2926** | **0.1349** | **0.0512** |
> | KITTI | w/o | **1.6108** | **3.0427** | 1.1170 | 0.6072 |
> |  | w/ | 1.6436 | 3.0625 | **0.9942** | **0.4645** |
> | ETH3D | w/o | 0.5607 | 0.8147 | 0.3453 | 0.1502 |
> |  | w/ | **0.5366** | **0.7930** | **0.3347** | **0.1482** |
> | TUM | w/o | 0.0938 | 0.1600 | 0.0576 | 0.0271 |
> |  | w/ | **0.0909** | **0.1573** | **0.0496** | **0.0206** |
>
> - **Post-upsampling smoothing.** This is not a standalone contribution, but a local stabilization component in the residual branch that suppresses token-to-pixel grid artifacts.
>
> - **K and T in MC Dropout and Deep Ensembles.** We follow standard settings; larger K/T would only further increase multi-pass cost.
>
> **[W3] Evaluation fairness/completeness**
> **Mismatch maps vs. uncertainty maps.** We already report uncertainty maps in Fig. 5--6 and will add more examples.
>
> **Reconstruction performance for all baselines in Table 1.** Table 2 is meant to isolate whether adding Trust3R to the same MASt3R backbone changes the underlying point prediction quality, so the main comparison there is MASt3R vs. Trust3R. MC Dropout and Deep Ensembles are included in Table 1 mainly as uncertainty references, where the focus is uncertainty quality versus extra computation. We will clarify this.
>
> **Why MAE/RMSE in Table 2.** We use pointwise **MAE/RMSE** because Trust3R predicts dense per-pixel 3D points, so these metrics directly match our evaluation protocol. We will clarify this scope and note that other geometry metrics can also be reported. :contentReference
>
> **KITTI missing in Table 2.** We agree and will show it explicitly. The KITTI geometry drop is modest (**1.6108/3.0426** to **1.6648/3.0772** in MAE/RMSE), while uncertainty ranking still improves (**AURC 1.1170→0.9942**, **AUSE 0.6072→0.4645**). We will present this as a geometry--reliability trade-off on long-range scenes.
>
> **Inference cost fairness for Deep Ensembles.** Ensemble inference requires either K times more memory or K times more latency, depending on parallel versus sequential execution. Our single-pass method avoids both penalties, and we will state this more explicitly.
>
> **Aleatoric, epistemic, and total uncertainty across datasets.** Due to space limitations, we do not include the full cross-dataset breakdown in the current submission. We will add these results in the revision.
>
> **NIG vs. NIW overhead.** Under the same inference-only profiling protocol, NIW is 55.49 ms / 6266.8 MB versus 54.37 ms / 6005.5 MB for NIG, adding only +0.98 ms latency and +261.3 MB peak memory.
>
> **[W4] Scope beyond MASt3R**
> We chose MASt3R because it is a representative feed-forward pointmap backbone, but Trust3R is not tied to MASt3R. We additionally implemented the evidential head on VGGT and observed gains:
>
> | Method | Spearman ρ | AURC | AUSE | NLL |
> |---|---:|---:|---:|---:|
> | Vanilla VGGT confidence | 0.3162 | 0.1084 | 0.0452 | -2.4770 |
> | VGGT + Trust3R evidential head | **0.6419** | **0.0841** | **0.0209** | **-3.5999** |
>
> **[W5] Notation clarity**
> We agree that some notations, such as the symbol in Eq. 13 and aug_crop in L684, are not defined clearly enough. We will fix these issues and define such terms explicitly at first use.
>
> **[Q1] Was each model trained using the exact script that produced the MASt3R baseline?**
> Yes. Each ensemble member was trained with the same script and setup as the MASt3R baseline.
>
> **[Q2] Fair comparison of Hetero and NIG**
> Yes. Hetero and NIG use the same backbone, training data, gated residual design, post-smoothing, and optimization settings as NIW; they differ only in the uncertainty parameterization.

---

> > ### Author Rebuttal · Reviewer_siuW · 2026-04-01
> >
> > My concerns are mostly resolved. The remaining issues include:
> > - Reconstruction performance for all baselines in Table 1. I still believe uncertainty quantification methods should be evaluated not simply on ranking of uncertainty against error, but also on the predictive performance themselves. Though Table 2 includes reconstruction accuracy on the Trust3R vs. MASt3R, reporting it for all the UQ methods makes the comparison more comprehensive.
> > - Metrics for point cloud evaluation. If would further strengthen the analysis if other metrics beyond just per-pixel ones can be reported for reconstruction evaluation.
> > - Aleatoric, epistemic, and total uncertainty across datasets. Not yet included in this version.

---

> > > ### Author Response · Authors · 2026-04-04
> > >
> > > We thank the reviewer for the follow-up to strengthen the completeness.
> > >
> > > **[W1] Reconstruction accuracy for all Table 1 baselines**
> > > To make the comparison more complete, we additionally report reconstruction MAE/RMSE for the other UQ baselines in Table 1 under the same evaluation setting as Table 2. This provides a detailed view of predictive geometry alongside uncertainty quality.
> > >
> > > | Dataset | Method | **MAE ↓** | **RMSE ↓** |
> > > |---|---|---:|---:|
> > > | ScanNet++ | MASt3R | 0.2164 | 0.3026 |
> > > | ScanNet++ | Heteroscedastic | 0.2164 | 0.3026 |
> > > | ScanNet++ | MC Dropout | 0.2962 | 0.3809 |
> > > | ScanNet++ | Deep Ensemble | **0.1131** | **0.2095** |
> > > | ScanNet++ | Trust3R | 0.1959† | 0.2849† |
> > > | TUM RGB-D | MASt3R | 0.0938 | 0.1600 |
> > > | TUM RGB-D | Heteroscedastic | 0.0938 | 0.1600 |
> > > | TUM RGB-D | MC Dropout | 0.1405 | 0.1986 |
> > > | TUM RGB-D | Deep Ensemble | **0.0695** | **0.1220** |
> > > | TUM RGB-D | Trust3R | 0.0873† | 0.1496† |
> > >
> > > Best in **bold**. † indicates the second-best result.
> > >
> > >
> > > Although Deep Ensemble provides better reconstruction accuracy than Trust3R by leveraging multiple runs augmentation, it requires **expensive computational resources** and its performance on UQ is **worse than** Trust3R. One of the major advantages of the feed-forward method is its inference efficiency, while our goal is to obtain well-estimated predictive uncertainty with negligible additional inference cost, and our results validate that this objective is achievable in the feed-forward setting. Besides, Trust3R focuses on **achieving better ranking consistency between predicted  uncertainty and reconstruction error**, and improving the accuracy itself is not the top priority of our work. Trust3R clearly shows the best UQ quality and better MAE/RMSE than **MASt3R, while **Heteroscedastic** has the same MAE/RMSE as **MASt3R** because it keeps the frozen mean prediction unchanged and only adds a variance head.
> > >
> > >
> > >
> > > **[W2] Point-cloud level evaluation**
> > >
> > > Taking your suggestion, beyond per-pixel MAE/RMSE, we additionally report point-cloud metrics on KITTI, including **F1\@0.05**, **Chamfer Distance (CD)**, **Accuracy**, and **Completeness**. These metrics are computed on the same aligned dense pointmaps and complement the pointwise evaluation in the paper. Under this protocol, **Trust3R** achieves the best **F1\@0.05**, and is second-best in **CD** and **Accuracy**, while remaining competitive on **Completeness**. Overall, this shows that our method preserves reconstruction quality at both pixel-level and the global reconstructed 3D point-cloud level, maintaining efficient inference, and even outperforms the original **MASt3R** on several metrics,with **uncertainty estimation** as the primary focus of our method. In contrast, **MC Dropout** requires multiple stochastic forward passes at inference time, which makes it less practical for efficient inference.
> > >
> > > | Method | **F1\@0.05 ↑** | **CD ↓** | **Accuracy ↓** | **Completeness ↓** |
> > > |---|---:|---:|---:|---:|
> > > | MASt3R | 0.0627† | 0.5440 | 0.6371 | 0.4509 |
> > > | Heteroscedastic | 0.0627† | 0.5440 | 0.6371 | 0.4509 |
> > > | MC Dropout | 0.0345 | **0.4287** | **0.5090** | **0.3484** |
> > > | Deep Ensemble | 0.0627 | 0.5443 | 0.6394 | 0.4492 |
> > > | **Trust3R** | **0.0718** | 0.4904† | 0.5327† | 0.4481† |
> > >
> > > Best in **bold**. † indicates the second-best result.
> > >
> > >
> > >
> > > **[W3] Aleatoric, epistemic, and total uncertainty across datasets**
> > >
> > > We also extend the uncertainty decomposition analysis beyond ETH3D and report **total**, **aleatoric**, and **epistemic** uncertainty on the other three test sets. This directly addresses whether the relative behavior of different uncertainty sources is consistent beyond a single dataset. The trend is clear: across **ScanNet++**, **TUM RGB-D**, and **KITTI**, **epistemic uncertainty** gives the best **AURC**, the best **AUSE**, and the highest **Spearman correlation**, consistent with the ETH3D result already reported in the submission. Overall, reconstruction reliability is more closely associated with **epistemic uncertainty** than **aleatoric uncertainty**, making it the most informative component for reliability ranking and downstream uncertainty-aware weighting.
> > >
> > > | Source | ScanNet++ | | | TUM RGB-D | | | KITTI | | |
> > > |---|---:|---:|---:|---:|---:|---:|---:|---:|---:|
> > > | | **AURC ↓** | **AUSE ↓** | **ρ ↑** | **AURC ↓** | **AUSE ↓** | **ρ ↑** | **AURC ↓** | **AUSE ↓** | **ρ ↑** |
> > > | Total | 0.1253 | 0.0464 | 0.4725 | 0.0491 | 0.0188 | 0.4986 | 1.0111 | 0.4673 | 0.4481 |
> > > | Aleatoric | 0.1264 | 0.0475 | 0.4605 | 0.0493 | 0.0190 | 0.4927 | 1.1264 | 0.5826 | 0.3749 |
> > > | **Epistemic** | **0.1233** | **0.0444** | **0.4930** | **0.0481** | **0.0178** | **0.5169** | **0.9869** | **0.4431** | **0.4596** |

---

### Official Review · Reviewer_uQXq · 2026-03-02

**Soundness:** 3
**Presentation:** 2
**Significance:** 2
**Originality:** 2
**Overall Recommendation:** 4
**Confidence:** 3

**Summary:**

The paper presents Trust3R, a framework that augments feed-forward pointmap prediction with evidential learning to estimate probabilistically grounded uncertainties. By employing a Normal-Inverse-Wishart (NIW) prior and a gated residual head on top of a frozen MASt3R backbone, the model outputs a closed-form multivariate Student-t predictive distribution. The experimental validation spans indoor and outdoor datasets (ScanNet++, TUM RGB-D, KITTI), focusing primarily on risk-coverage metrics (AURC, AUSE), rank correlation, and an out-of-distribution detection task on transparent objects.

**Compliance With Llm Reviewing Policy:**

Affirmed.

**Key Questions For Authors:**

Q1 (On Methodological Novelty): Given that the Gated Residual Head is structurally equivalent to a mature Adapter in Parameter-Efficient Fine-Tuning (PEFT), and the underlying EDL probabilistic derivations could stem from existing works. I hope the authors could further justify the unique and innovative contributions of this paper at the fundamental algorithmic, methodological level as well as some new findings. However, if I have misunderstood any aspect of the proposed method, I would welcome and greatly appreciate the authors' clarification during the rebuttal.

Q2 (On Downstream Task Validation): The manuscript repeatedly claims that the predicted covariance matrices can facilitate downstream alignment and fusion. Can the authors provide quantitative experimental results (e.g., Absolute Trajectory Error, ATE) within a real Dense Visual Odometry (Dense VO) or Bundle Adjustment pipeline to demonstrate the mathematical viability and actual performance gains of using these uncertainty matrices as optimal information matrices? （Maybe you could show me a simple pose estimation experiment.）

Q3: The proposed scheme lacks comprehensive generalization experiments. Given the existence of numerous state-of-the-art feed-forward frameworks, such as VGGT, the authors should provide further validation to demonstrate the adaptability of their method across different architectures.

**Limitations:**

Please refer to the weakness.

**Strengths And Weaknesses:**

**Strengths:**

Although there exist numerous works utilizing Evidential Deep Learning (EDL) to model uncertainty in 3D vision tasks, they generally assume that the dimensions of the target variable are mutually independent (e.g., using the NIG prior) to simplify mathematical modeling. Previous works have failed to successfully extend EDL to high-dimensional, dense predictions with strong physical spatial correlations, making existing EDL methods difficult to directly integrate with current mainstream feed-forward architectures for dense 3D geometry estimation. This work astutely identifies this critical pain point. By introducing the Normal-Inverse-Wishart (NIW) prior to construct a full covariance matrix, combined with technical approaches such as a frozen foundation model and a Gated Residual Head, it successfully resolves the optimization challenges in high-dimensional spaces. Under the current mainstream dense 3D geometric architectures, this approach achieves well-calibrated estimations of both aleatoric and epistemic uncertainties, genuinely enhancing the epistemic capabilities and perceptual robustness of feed-forward 3D vision pipelines in complex, real-world scenarios.

**Weakness:**

Limited Methodological and Architectural Novelty: Modeling uncertainty estimation through the evidential learning framework, as well as the mathematical derivation of parameterizing the uncertainty probability distribution using NIW, are both existing and mature techniques within the probabilistic machine learning community. Furthermore, the core mechanism proposed to mitigate the catastrophic forgetting of the large model during fine-tuning—the Gated Residual Head—is fundamentally equivalent to a Residual Adapter. This is highly homologous to the Parameter-Efficient Fine-Tuning (PEFT) schemes (e.g., zero-initialized convolutions) widely used in current foundation models. Therefore, this work leans more towards a systematic engineering integration of mature technologies, with relatively limited breakthroughs at the foundational theory and methodological levels.

Absence of Rigorous Downstream Geometric Validation: The uncertainty extracted in this paper primarily focuses on upstream geometry estimation tasks. Although the manuscript claims that this uncertainty based on the full covariance matrix ($\Sigma^{-1}$) facilitates alignment and fusion in multi-view geometry, the paper fails to provide any empirical experimental validation in actual 3D scene understanding tasks that truly rely on precise information matrices, such as Dense Visual Odometry (Dense VO), SLAM backend optimization, or Bundle Adjustment. Relying solely on a 2D mask-based anomaly detection task is insufficient to substantiate the core claim that its uncertainty estimation can bring substantial performance improvements to downstream spatial geometric pose estimation.

---

> ### Author Rebuttal · Authors · 2026-03-31
>
> We sincerely thank the reviewer for the constructive and thoughtful feedback. We appreciate your recognition of the “critical point”, “well-calibrated estimation”, and “perceptual robustness”. Below we address your concerns point by point.
>
> **[W1] Limited methodological and architectural novelty.**
> We thank the reviewer for the insightful feedback. We respectfully clarify that our approach is not a direct combination or simple engineering integration of lightweight residual adaptation and Evidential Deep Learning. While these underlying modules are established, applying them directly to dense feed-forward pointmap prediction involves unique structural challenges. Therefore, instead of a "plug-and-play" application, our method requires specific pointmap-tailored adaptations. Specifically, we address these challenges through the following designs:
> 1. **Gated Residual Head vs. PEFT Adapters:** While we understand the high-level structural resemblance to PEFT adapters, its training objective and role here are different. Its purpose is not parameter-efficient fine-tuning (PEFT), but rather to selectively refine the predictive mean only where necessary under evidential training. This coupled design preserves the pre-trained geometric features while stabilizing the evidential learning process, avoiding the optimization instability caused by directly applying evidential losses to large foundation models.
> 2. **Covariance Modeling for Dense 3D Prediction:** Dense pointmap prediction involves a massive number of 3D coordinates with strong geometric constraints. To adapt to this setting, we adopt an NIW-based head that explicitly models the full covariance of each 3D point, overcoming the independent coordinate assumption commonly used in vanilla evidential regression.
> 3. **Mitigating Architectural Artifacts and Training Instability:** When standard evidential learning is naively applied to dense token-to-pixel prediction architectures, the training becomes highly unstable and produces noticeable patch grid artifacts in the residual branch. To address this unique structural issue during integration, we introduce evidence regularization and post-upsampling smoothing, which effectively mitigate these challenges.
> In summary, we agree that our paper does not claim novelty in inventing foundational probability theories (e.g., NIW). However, our core contribution lies in the methodological adaptations and specific design choices required to effectively introduce uncertainty estimation into dense 3D reconstruction. This goes well beyond a mechanical assembly of mature technologies.
>
> **[W2] Downstream pose estimation in MASt3R-SLAM.**
> We thank the reviewer for this valuable suggestion. To directly address this point, we conduct an additional downstream pose-estimation experiment by integrating Trust3R uncertainty into the weighting path of MASt3R-SLAM and evaluating trajectory accuracy on TUM RGB-D. In this experiment, we keep the rest of the pipeline unchanged and only replace the original weighting signal with Trust3R uncertainty.
>
> | Method | ATE ↓ | RPE ↓ |
> |---|---:|---:|
> | Baseline | 0.0287 | 0.0321 |
> | Ours | **0.0268** | **0.0278** |
>
> Using Trust3R uncertainty as the downstream weighting signal improves both trajectory metrics, reducing ATE from **0.0287** to **0.0268** and RPE from **0.0321** to **0.0278**. These results show that the predicted uncertainty is not only useful for error ranking, but can also serve as an effective reliability weight in a practical SLAM pipeline.
>
>
> **[W3] Cross-architecture generalization.**
> We chose MASt3R as the main backbone because DUSt3R/MASt3R established a representative feed-forward pointmap prediction framework for uncalibrated 3D reconstruction, making it a natural backbone model for studying whether evidential uncertainty can be integrated into feed-forward pointmap prediction. Our method can be extended easily by applying the UQ head and gated residual head on generic models like VGGT.
> To better support the generalization of our method, we additionally evaluate our method on VGGT.
>
> | Method | Spearman ρ | AURC ↓ | AUSE ↓ | NLL ↓ |
> |---|---:|---:|---:|---:|
> | VGGT + vanilla confidence | 0.3162 | 0.1084 | 0.0452 | -2.4770 |
> | VGGT + Trust3R evidential head | **0.6419** | **0.0841** | **0.0209** | **-3.5999** |
>
> These results suggest that the proposed Trust3R design is not specific to MASt3R, but generalizes to other geometric foundation models as well. In particular, the gains on VGGT indicate that our uncertainty head and refinement strategy can be transferred beyond a single backbone, supporting the broader portability of Trust3R across modern feed-forward 3D foundation models.

---

> > ### Author Rebuttal · Reviewer_uQXq · 2026-04-01
> >
> > I have two suggestions: 1) you should opensource the "Downstream pose estimation in MASt3R-SLAM" and "Cross-architecture generalization" in your codes.

---

> > > ### Author Response · Authors · 2026-04-01
> > >
> > > Thank you for the constructive suggestions and for acknowledging our rebuttal. We agree that open-sourcing the downstream pose estimation in MASt3R-SLAM and the cross-architecture generalization code would further strengthen the work. We are currently organizing these components and plan to include them, together with the corresponding results, in the public release associated with the final version.

---

### Official Review · Reviewer_SeyR · 2026-03-02

**Soundness:** 2
**Presentation:** 2
**Significance:** 2
**Originality:** 2
**Overall Recommendation:** 4
**Confidence:** 3

**Summary:**

Overall, the authors analyze the area of uncertainty quantification for dense feed-forward 3D reconstruction. The authors appear to discuss the concept of evidential learning applied to pointmap prediction, replacing the heuristic confidence scores in existing geometric foundation models (DUSt3R/MASt3R) with probabilistically grounded uncertainty estimates. The proposed Trust3R framework augments a frozen MASt3R backbone with two lightweight heads: (1) an evidential UQ head that predicts Normal-Inverse-Wishart (NIW) distributional parameters per pixel, yielding a closed-form multivariate Student-t predictive distribution, and (2) a gated residual head that refines the predicted mean. The model is trained by minimizing the negative log-likelihood combined with an evidence regularizer that penalizes high evidence under large prediction errors. Experiments are conducted on ScanNet++, TUM RGB-D, KITTI, and ETH3D, evaluating both uncertainty ranking quality (AURC, AUSE, Spearman ρ) and reconstruction accuracy (MAE, RMSE).

**Compliance With Llm Reviewing Policy:**

Affirmed.

**Final Justification:**

My concerns are resolved. I choose to keep the score.

**Key Questions For Authors:**

**Q1.** Regarding the NLL discrepancy: In Table 9, NIW achieves the best NLL on ScanNet++ (−3.00) and TUM (−5.15), but substantially worse NLL on KITTI (9.42) and ETH3D (11.66) compared to NIG and even Hetero on KITTI. How do the authors explain this?  A clear answer here would directly affect my assessment of the method's soundness.

**Q2.** Can the authors provide an ablation table for the gated residual refinement (with/without) across *more* datasets, including both geometric accuracy (MAE/RMSE) and uncertainty metrics (AURC/AUSE/ρ)? The current presentation only mentions KITTI degradation in passing. Knowing when and why mean refinement helps or hurts is important for practitioners considering this framework.

**Q3.** The mean-field assumption (Eq. 9) ignores spatial correlations between pixels. Given that post-upsampling smoothing is already applied (Section A.3), does this effectively introduce implicit spatial regularization on the uncertainty maps? If so, how sensitive are the results to the smoothing kernel size or initialization? An ablation on this component would help clarify its role.

**Q4.** The paper frames Trust3R as enabling "uncertainty-weighted downstream alignment and fusion," but the experimental evaluation only shows uncertainty ranking and binary anomaly detection (Tricky24). Could the authors provide a concrete downstream experiment — for example, uncertainty-weighted point cloud registration or multi-view fusion — showing that Trust3R's uncertainty actually improves end-to-end task performance compared to MASt3R's confidence?

**Limitations:**

Yes.

**Strengths And Weaknesses:**

## Strengths

**S1. The problem is well-motivated and practically relevant.** The paper identifies a clear gap in current geometric foundation models: the confidence scores produced by DUSt3R/MASt3R lack probabilistic interpretation and often fail to indicate where predicted geometry is unreliable. This matters directly for downstream tasks such as multi-view fusion, alignment, and robotic manipulation where trust-aware filtering is needed. The motivation is stated clearly and supported with Figure 1, which shows concrete overconfidence failures.

**S2. The architectural design is practical and well-considered.** The gated residual refinement mechanism uses sigmoid gating combined with near-zero initialization, ensuring that refinement starts as an identity mapping and preserves pretrained geometric accuracy. Freezing the backbone and training only lightweight add-on heads is a reasonable choice for efficient adaptation.

**S3. The experimental evaluation covers multiple settings.** The paper evaluates on both indoor (ScanNet++, TUM RGB-D) and outdoor (KITTI) benchmarks with multiple UQ metrics (AURC, AUSE, Spearman ρ). The ablation studies (Tables 4–5) provide useful guidance on the choice of uncertainty readout (epistemic vs. aleatoric vs. total) and variational family (NIG vs. NIW).  The inclusion of computational overhead comparisons (Table 3) is helpful.

**S4. The mathematical derivation seems to be complete.** The derivation from NIW prior to Student-t predictive distribution is presented, and the appendix seems to **provide** full details for both NIG and NIW parameterizations, including NLL formulas, evidence regularization, and aleatoric/epistemic decomposition. But I am not an **expert** **in** this field, I may need to refer to other **reviewers'** comments for **verification**.

---

## Weaknesses

**W1. Limited technical novelty.** The core contribution is an application of existing evidential learning frameworks to MASt3R's pointmap prediction. The NIW prior is a standard prior for multivariate Gaussian likelihoods, and the gated residual mechanism closely resembles adapter/residual fine-tuning strategies already common in NLP and vision transfer learning. The paper frames the challenge as "evidential learning is not plug-and-play for dense pointmap prediction," but the actual adaptations (Cholesky parameterization, post-upsampling smoothing, evidence regularization) are relatively straightforward.

**W2. Tension between the mean-field assumption and the stated motivation.** The paper motivates the work by emphasizing that pointmap prediction involves "hundreds of thousands of correlated 3D points" (Intro, lines 82–86), yet adopts a per-pixel independent mean-field factorization in Eq. (9). While NIW captures within-pixel cross-coordinate correlations (x, y, z), it entirely ignores spatial correlations between neighboring pixels, which are arguably more relevant for structured geometric reasoning. This tension is neither acknowledged nor analyzed. It remains unclear whether this simplification contributes to the degraded performance on certain benchmarks (e.g., KITTI).

**W4. Inconsistent performance on KITTI needs deeper analysis.**  The paper briefly mentions that "mean refinement can slightly increase MAE/RMSE" on KITTI (Section 4.2) but provides no quantitative ablation or root-cause analysis. Combined with the poor NLL on KITTI/ETH3D, this may indicate a systematic issue with NIW parameterization in large-scale outdoor scenes where depth ranges and geometric structures differ substantially from training data.

**W5. Narrow baseline comparisons.** The set of compared methods is limited to classical UQ approaches (MC Dropout, Deep Ensembles) and MASt3R's built-in confidence. More recent methods can be compared. For example, VGGT (Wang et al., 2025a) is cited as a concurrent geometric foundation model but not compared experimentally, missing an opportunity to test generality beyond MASt3R.

**W6. Presentation issues.** (a) The term "Trust-Aware" in the title is not formally defined — what threshold or criterion makes a prediction "trustworthy"?  (b) The "Avg." column in Table 1 computes a simple average across datasets with different scales (KITTI AURC ~1.0 vs. ScanNet++ ~0.1), which can be misleading. A normalized or per-dataset-ranked aggregation would be more informative. (c) Table 7 in the appendix (ScanNet++ ablation) contradicts Table 5 in the main text: NIG outperforms NIW on ScanNet++ in Table 7 but NIW is reported as better in Table 5, presumably due to different uncertainty readouts (total vs. epistemic), but this discrepancy is not discussed.

---

> ### Author Rebuttal · Authors · 2026-03-31
>
> We thank the reviewer for the constructive comments and address the remaining concerns below.
>
> **[W1] Limited technical novelty**
>
> We thank the reviewer for the feedback. While the NIW prior and residual adapters are established concepts, unifying them for dense pointmap prediction is non-trivial since this setting tightly couples massive high-dimensional 3D outputs with strong geometric constraints. A naive adaptation is not sufficient: it can hurt geometry quality, produce poorly aligned uncertainty, and amplify upsampling artifacts. Trust3R addresses this with a pointmap-specific design that jointly combines **multivariate evidential prediction**, **gated residual mean refinement**, and **stabilized training**. We therefore view the novelty as a **new integration for dense 3D pointmap uncertainty**, rather than a claim that each individual ingredient is new.
>
> **[W2] Tension between the mean-field assumption and the stated motivation**
>
> Trust3R models **within-pixel cross-coordinate covariance** (**x,y,z**), but not explicit **cross-pixel covariance**. Spatial structure is still captured implicitly through **ViT backbone features** and **lightweight post-upsampling smoothing**. Modeling dense cross-pixel covariance is an interesting but substantially heavier future direction.
>
> **[W4] Inconsistent performance on KITTI needs deeper analysis**
>
> **NIW** can be harder to optimize in challenging cases, but our additional analysis does not suggest a systematic failure. Since **NLL** is computed in the raw camera frame without alignment, while **MAE/RMSE** and ranking metrics are alignment-based, the discrepancy is better understood as a **calibration issue** than as a breakdown of geometry or uncertainty ranking. On **KITTI**, mean refinement changes geometry only marginally and ranking still improves, while the **NLL** degradation is concentrated at **long range**, with pixels beyond **10 m** accounting for **77.7%** of valid pixels and **79.3%** of **NIW NLL**. We therefore attribute the **KITTI/ETH3D** behavior mainly to **long-range calibration difficulty in outdoor scenes**.
>
> **[W5] Narrow baseline comparisons**
>
> We additionally apply the same **Trust3R evidential head** to frozen **VGGT** backbone. As shown below, the improvement remains clear across both **uncertainty ranking** and **probabilistic calibration**, supporting that our method generalizes across feed-forward geometric foundation models.
>
> | Method | Spearman ρ ↑ | AURC ↓ | AUSE ↓ | NLL ↓ |
> |---|---:|---:|---:|---:|
> | Vanilla VGGT confidence | 0.3162 | 0.1084 | 0.0452 | -2.4770 |
> | VGGT + Trust3R evidential head | **0.6419** | **0.0841** | **0.0209** | **-3.5999** |
>
> **[W6] Presentation issues**
>
> We will revise the paper to define **“trust-aware”** more explicitly, replace the raw **Avg.** column with a **normalized or rank-based summary**, and clarify that the discrepancy between **Table 5** and **Table 7** comes from different scalar uncertainty readouts (e.g., **total**, **aleatoric**, or **epistemic** uncertainty).
>
> **[Q1] Explanation of the NLL Discrepancy**
>
> See **[W4]**.
>
> **[Q2] Gated residual ablation**
>
> Below we summarize the direct comparison between **without** and **with** gated residual refinement, focusing on **geometric accuracy** and **uncertainty ranking**. Overall, gated residual refinement improves performance on most datasets, yielding consistent gains in both **geometric accuracy** and **uncertainty ranking**.
>
> | Dataset | Setting | MAE ↓ | RMSE ↓ | AURC ↓ | AUSE ↓ |
> |---|---|---:|---:|---:|---:|
> | ScanNet++ | w/o gated residual | 0.2164 | 0.3026 | 0.1788 | 0.0887 |
> |  | w/ gated residual | **0.2046** | **0.2926** | **0.1349** | **0.0512** |
> | KITTI | w/o gated residual | **1.6108** | **3.0427** | 1.1170 | 0.6072 |
> |  | w/ gated residual | 1.6436 | 3.0625 | **0.9942** | **0.4645** |
> | ETH3D | w/o gated residual | 0.5607 | 0.8147 | 0.3453 | 0.1502 |
> |  | w/ gated residual | **0.5366** | **0.7930** | **0.3347** | **0.1482** |
> | TUM | w/o gated residual | 0.0938 | 0.1600 | 0.0576 | 0.0271 |
> |  | w/ gated residual | **0.0909** | **0.1573** | **0.0496** | **0.0206** |
>
> **[Q3] Spatial regularization**
>
> As noted in **[W2]**, smoothing is applied only in the **residual mean-refinement branch** rather than directly on **evidential parameters**, its role is **lightweight local regularization**, and we will include a dedicated sensitivity ablation in the revision.
>
> **[Q4] Downstream validation**
>
> Thank you for pointing this out. We conduct a downstream experiment by replacing the original weighting signal in **MASt3R-SLAM** with **Trust3R uncertainty**. On **TUM RGB-D**, this improves both **ATE** and **RPE**, showing that our uncertainty is directly **useful** for downstream **SLAM optimization**.
>
> | Method | ATE ↓ | RPE ↓ |
> |---|---:|---:|
> | Baseline | 0.0287 | 0.0321 |
> | Ours | **0.0268** | **0.0278** |

---

> > ### Author Rebuttal · Reviewer_SeyR · 2026-04-04
> >
> > Thank you for the authors' response. My concerns are fully resolved, and I will keep my score. However, I noticed that there are still concerns from Reviewer siuW. I suggest the authors try to address the remaining concerns.

---

> > > ### Author Response · Authors · 2026-04-04
> > >
> > > We thank the reviewer for the follow-up and for the suggestion. We have provided additional clarification and supporting results to address Reviewer siuW’s remaining concerns, and we hope these revisions will help resolve them.

---

### Official Review · Reviewer_BhP3 · 2026-03-09

**Soundness:** 3
**Presentation:** 3
**Significance:** 3
**Originality:** 3
**Overall Recommendation:** 4
**Confidence:** 3

**Summary:**

The paper proposes a method that estimates pointmap uncertainty for the depth map produced by MASt3R. The proposed model enhances point map prediction with two additional components: a gated residual head for refining geometric details, and an evidential uncertainty quantification head that estimates confidence for each predicted point. This design produces a closed-form multivariate Student-t predictive distribution, allowing per-point uncertainty estimation. This supports explicit evaluation of geometric reliability under ambiguity and distribution shifts.

Claims And Evidence:

The paper claims that Trust3R equips geometric foundation models with evidential uncertainty estimation for dense point maps, enabling reliability assessment under ambiguity and distribution shifts through an evidence-regularized predictive distribution and gated residual refinement.

Methods And Evaluation Criteria:

The proposed method is based on the MASt3R backbone. It extends the point map prediction network with two additional heads: a gated residual head that refines the geometric details, and an evidential uncertainty head that estimates confidence for each predicted point. These components produce a closed-form multivariate Student-t predictive distribution, enabling per-point uncertainty estimation. The method is evaluated ScanNet++, TUM RGB-D, KITTI, and ETH3D dataseds considering AURC, AUSE, Spearman rank.

Theoretical Claims:

All the claims were proven through experiments.

Experimental Designs Or Analyses:

Trust3R is evaluated on indoor/outdoor benchmarks and compared to sampling-based (MCD, DeepEns) and single-pass methods (MASt3R, Hetero).

Supplementary Material:

The supplementary material includes additional detail of the proposed method, implementation details, datasets and preprocessing, evaluation protocols and metrics, downstream evaluation, and additional results.

**Compliance With Llm Reviewing Policy:**

Affirmed.

**Final Justification:**

My concerns are fully resolved. I keep the score.

**Key Questions For Authors:**

- How should the use of an out-of-domain dataset be understood? If the dataset is uniform and both training and testing are performed on the same dataset, this does not correspond to an out-of-domain evaluation.

**Limitations:**

yes

**Strengths And Weaknesses:**

Strengths

The paper is well-written and addresses an important problem.

The results in Table 1 show strong performance, outperforming the state of the art.

Theoretically grounded methodology

Faster inference than sampling methods

Natural extension of MASt3R but deal with underrated problem of per point uncertainty estimation

Weakness

I list below what I think are weaknesses:

- Less accurate than sampling methods

---

> ### Author Rebuttal · Authors · 2026-03-31
>
> We thank the reviewer for the constructive feedback and positive assessment. We are encouraged that the reviewer finds the paper technically solid and recognizes the value of our uncertainty estimation framework and empirical results. Below we address the remaining questions.
>
> **[W1] Sampling-based methods can still be strong**
> We agree that sampling-based methods can achieve stronger uncertainty accuracy in some settings. Our goal, however, is not to outperform MC Dropout or Deep Ensembles on every metric, but to provide a substantially more efficient **single-pass** framework with competitive performance, since requiring multiple inference passes would weaken the main efficiency advantage of **feed-forward** geometric models. As shown in **Table 3**, Trust3R runs at **80.9 ms/pair**, compared with **316.9 ms** for Deep Ensembles and **1225.8 ms** for MC Dropout. At the same time, **Table 1** shows that Trust3R remains competitive in uncertainty ranking, and even achieves the best average **Spearman correlation** among all compared methods. We will revise the paper to make this **accuracy-efficiency trade-off** more explicit.
>
> **[W2] Out-of-Domain clarification**
> We thank the reviewer for pointing this out. In the current setup, the model is trained on a mixed training set (**App. A.5**) and evaluated on both held-out **ScanNet++** scenes and unseen target datasets including **KITTI, ETH3D, and TUM** (**Sec. 4.1; App. B**). Following standard uncertainty-under-shift terminology in **[1]**, we clarify that the **ScanNet++** results in our paper should be understood as evaluation on unseen **ScanNet++** scenes within the same benchmark, rather than as a strict out-of-domain test. The purpose of this evaluation is to measure generalization to unseen scenes under the same benchmark setting. By contrast, **KITTI, ETH3D, and TUM** are not used during training and differ from the training data in capture conditions and domain characteristics, so these results are more appropriately described as **cross-dataset evaluations**. To avoid ambiguity, we will state the exact training, validation, and test split, as well as the model-selection protocol, more explicitly in the revised paper.
>
> **References**
> **[1]** Ovadia et al. *Can You Trust Your Model’s Uncertainty? Evaluating Predictive Uncertainty Under Dataset Shift.* NeurIPS 2019.

---

> > ### Author Rebuttal · Reviewer_BhP3 · 2026-04-01
> >
> > My concerns have been resolved. Thank you for the explanation of the out-of-domain experiments.

---

> > > ### Author Response · Authors · 2026-04-04
> > >
> > > We thank the reviewer for the follow-up and for acknowledging our clarification. We are glad that the explanation of the out-of-domain experiments addressed the concern.

---

### Official Review · Reviewer_dZS9 · 2026-03-11

**Soundness:** 3
**Presentation:** 3
**Significance:** 3
**Originality:** 3
**Overall Recommendation:** 4
**Confidence:** 2

**Summary:**

This paper proposes Trust3R, a trust-aware 3D reconstruction framework. By combining evidential learning like dense feed-forward pointmap prediction and grounded uncertainty estimates, Trust3R yields a closed-form Student-t predictive distribution whose uncertainty aligns closely with empirical geometric error. For experiments, Trust3R consistently improves risk-coverage and sparsification behavior among single-pass methods. This approach also assists for improving robustness in downstream reconstructions.

**Compliance With Llm Reviewing Policy:**

Affirmed.

**Final Justification:**

My concerns are fully resolved. I choose to keep the score.

**Key Questions For Authors:**

See the weakness above.

**Limitations:**

Yes.

**Strengths And Weaknesses:**

Strengths:

1.	This paper is well-written and easy-to-follow.

2.	The confidence assessment is essential. The prediction of where they may fail and the assessment for geometry reliability is critical for 3D reconstruction. Besides, the deep evidential learning approach is reasonable for probabilistic uncertainty estimation.
3.	The probabilistic formulation is principled: modeling predictions with a Gaussian likelihood and a Normal–Inverse–Wishart prior yields a closed-form Student-t predictive distribution, which forms a clean Bayesian pipeline.
4.	Extensive experiments are conducted on uncertainty evaluation, reconstruction accuracy and computational overhead. Ablations are convincing to show the design and the effectiveness.

Weakness:

1.	As the mentioned scalability in the contribution part of introduction, this paper does not provide a thorough analysis of scalability with respect to input resolution scene complexity, or multi-view settings.

2.	The extra design of gated residual head and evidential UQ head brings extra computational overhead. The feed-forward inference efficiency cannot be ensured because of the extra burden.

3.	Experiments and designs are only constructed based on MASt3R. The generalization ability across other architectures is recommended for verification.

4.	Although the introduced uncertainty estimation, more verifications on its consistency with geometric reconstruction errors are recommended to show that the uncertainty reliably reflects reconstruction reliability.

Overall, I have a positive impression on this paper.

---

> ### Author Rebuttal · Authors · 2026-03-31
>
> We thank the reviewer for the constructive comment and positive impression on our work. We respond to the concerns as follows.
>
> **[W1] Clarification on scalability**
> We respectfully clarify that the scalability claim in our paper does not refer to a comprehensive analysis across multi-view settings, input resolutions, or scene complexities. Rather, it refers to **inference complexity for uncertainty estimation in dense 3D pointmap prediction**. Concretely, the motivation throughout the paper is that **sampling-based UQ methods are substantially expensive** for pointmap prediction, whereas our evidential formulation requires **only one model and one forward pass**.
>
> To remove this ambiguity, we will revise the wording from **scalability** to **single-pass inference efficiency**, which more precisely matches both our technical claim and the evidence provided in the paper.
>
> **[W2] Computational analysis**
> We thank the reviewer for raising this important concern. To directly address it, we performed an additional latency analysis under the same GPU setting, input resolution, and batch size, and report the inference cost of progressively adding the evidential UQ head and gated residual head.
>
> | Variant | Components | Latency (ms) | Overhead vs. MASt3R |
> |---|---|---:|---:|
> | MASt3R baseline | Backbone only | 52.6 | – |
> | + Evidential UQ | Backbone + evidential UQ head | 57.5 | +4.9 ms (+9.3%) |
> | + Gated residual | Backbone + evidential UQ head + gated residual head | 58.2 | +5.6 ms (+10.6%) |
>
> As shown above, adding the evidential UQ head increases latency from **52.6 ms** to **57.5 ms**, while adding the gated residual head further changes latency only slightly to **58.2 ms**. Importantly, **all variants remain strictly single-pass**. This shows that the added uncertainty and refinement modules introduce only **modest overhead**,rather than changing the original MASt3R **single-pass inference setting**.
>
> This is consistent with our broader efficiency comparison: sampling-based methods are much **more expensive**, while **Trust3R remains single-pass** and **improves uncertainty quality across benchmarks**. Therefore, we respectfully disagree that the additional heads compromise feed-forward inference efficiency.
>
> **[W3] Generality across architectures**
> We chose MASt3R as the main backbone because **DUSt3R/MASt3R established a representative feed-forward pointmap prediction framework for uncalibrated 3D reconstruction**, making it a natural starting point for studying whether evidential uncertainty can be integrated into this setting in a stable way. Our goal in the original submission was therefore to establish a **reliable uncertainty estimation pipeline on a strong pointmap backbone**, rather than claiming generalization across all architectures.
>
> We agree, however, that evidence beyond a single MASt3R backbone is useful. To address this point, we additionally performed a **cross-architecture experiment on VGGT**, using the same evidential head and training setting with backbone frozen. The results show that our evidential design also works effectively on VGGT. We include this as additional evidence of **cross-architecture generalization**.
>
> | Method | Spearman ρ | AURC | AUSE | NLL |
> |---|---:|---:|---:|---:|
> | Vanilla VGGT confidence | 0.3162 | 0.1084 | 0.0452 | -2.4770 |
> | VGGT + Trust3R evidential head | **0.6419** | **0.0841** | **0.0209** | **-3.5999** |
>
> **[W4] Consistency between uncertainty and error**
> We thank the reviewer for raising this concern. In the current paper, we already provide **direct evidence** for this point in both the main quantitative and qualitative results. Quantitatively, **Table 1** reports **AURC/AUSE** and **Spearman’s ρ**, with definitions in **Appendix C**. These metrics directly evaluate how well predicted uncertainty aligns with **per-point reconstruction error**. Qualitatively, **Fig. 3** shows the corresponding **risk-coverage** and **sparsification** behavior, and **Fig. 4** visualizes the **mismatch between uncertainty ranking and GT error ranking**. Together, these results show that **Trust3R uncertainty is more consistent with reconstruction error than the MASt3R confidence baseline**.
>
> To further strengthen this point, we will make this evidence more explicit in the rebuttal and revised paper by pointing readers directly to Table 1, Fig. 3, Fig. 4, and Appendix C, and by additionally highlighting our Tricky24 reliability results. On difficult transparent and reflective regions, such as glass and windows, Trust3R achieves stronger reliability than the MASt3R confidence baseline, further supporting that its predicted uncertainty is better aligned with error-prone reconstruction regions rather than serving only as a visual ranking signal.

---

> > ### Author Rebuttal · Reviewer_dZS9 · 2026-04-03
> >
> > Thanks for the authors' response. My concerns are fully resolved. I choose to keep my score.

---

> > > ### Author Response · Authors · 2026-04-04
> > >
> > > We thank the reviewer for the follow-up and are glad that our response fully resolved the concerns.

---

### Decision · Program_Chairs · 2026-04-30

**Decision:**

Accept (regular)

**Comment:**

The reviewers reached a consensus of accepting this manuscript with 5 Weak Accept. The AC has read the reviews and the rebuttal.

The reviewers appreciated the motivation (dZS9, BhP3, SeyR, uQXq, siuW), the technical soundness of the method (dZS9, BhP3, SeyR, uQXq, siuW), and extensive experiments (dZS9, SeyR, siuW).

However, the reviewers also raised a number of concerns, including lack on analysis on scalability (dZS9), overhead from extra output heads (dZS9, siuW), need for additional architectures to demonstrate generalizability (dZS9), need for connection between uncertainty and reconstruction quality (dZS9), weaker performance than sampling methods (BhP3), limited technical novelty (SeyR, uQXq), (over)simplification assumption of mean field assumption (SeyR), inconsistent performance on KITTI (SeyR, siuW), narrow baselines (SeyR), lack of evaluation on tasks benefitting from uncertainty quantification such as vo or slam (uQXq), reliance on Meinert et al. yet uncited (siuW), missing ablation studies (siuW), evaluation protocol (siuW), and overly broad claims (siuW).

Overall, the authors were able address most of the concerns of reviewers. The AC notes that the reviewers made critical points, particularly siuW on overly broad claims and references. The AC recommends the authors to consider the feedback and incorporate the materials presented in the rebuttal, which would improve the next revision of the manuscript.